# Improved Environmental Stimulus and Biological Competition Tactics Interactive Artificial Ecological Optimization Algorithm for Clustering

**DOI:** 10.3390/biomimetics8020242

**Published:** 2023-06-07

**Authors:** Wenyan Guo, Mingfei Wu, Fang Dai, Yufan Qiang

**Affiliations:** School of Science, Xi’an University of Technology, Xi’an 710054, China; ryan2258@163.com (M.W.); daifang@xaut.edu.cn (F.D.); q1332609463@163.com (Y.Q.)

**Keywords:** artificial ecological optimization algorithm, environmental stimulus, population diversity, competition mechanism, engineering optimization, K-means clustering

## Abstract

An interactive artificial ecological optimization algorithm (SIAEO) based on environmental stimulus and a competition mechanism was devised to find the solution to a complex calculation, which can often become bogged down in local optimum because of the sequential execution of consumption and decomposition stages in the artificial ecological optimization algorithm. Firstly, the environmental stimulus defined by population diversity makes the population interactively execute the consumption operator and decomposition operator to abate the inhomogeneity of the algorithm. Secondly, the three different types of predation modes in the consumption stage were regarded as three different tasks, and the task execution mode was determined by the maximum cumulative success rate of each individual task execution. Furthermore, the biological competition operator is recommended to modify the regeneration strategy so that the SIAEO algorithm can provide consideration to the exploitation in the exploration stage, break the equal probability execution mode of the AEO, and promote the competition among operators. Finally, the stochastic mean suppression alternation exploitation problem is introduced in the later exploitation process of the algorithm, which can tremendously heighten the SIAEO algorithm to run away the local optimum. A comparison between SIAEO and other improved algorithms is performed on the CEC2017 and CEC2019 test set.

## 1. Introduction

In the realm of optimization, the emerging heuristic algorithm has become a focus of researchers due to its convenience and comprehensibility. Swarm intelligence (SI) is an important branch of biological heuristic computing. The SI algorithm takes advantage of the collaborative competitive relationship among populations to pilot the population to conduct the optimal solution, which fascinates a great deal of investigators. Several novel algorithms have been put forward in recent years. For example, SFO [1], HHO [2], BMO [3], EJS [4], MCSA [5], SDABWO [6], STOA [7], SCA [8], SSA [9] et al. Swarm-based intelligent optimization algorithm has greatly promoted its application in engineering optimization because it has abandoned gradient information in traditional optimization algorithms, such as time series prediction [10], image segmentation [11], feature selection [12] and cloud computing scheduling tasks [13].

The AEO algorithm [14] is a SI algorithm introduced by Zhao et al. in 2019. The AEO algorithm is designed according to energy flow and material circulation in natural ecosystem. The algorithm model is established through three stages of production, consumption and decomposition. The first stage is the producer, who itself obtains energy from nature and uses it to describe the plants in the ecosystem. The second stage is consumer, whose main body is an animal; the final stage is the decomposer, which feeds on both the producer and the consumer.

In the population of AEO algorithm, all individuals are called consumers except for one producer and decomposer. Consumers can be divided into herbivorous animals, carnivorous animals and omnivorous animals according to their different ways of foraging. Energy levels in an ecosystem decrease along the food chain, so producers have the highest energy levels, and decomposers have the lowest. AEO algorithm has a strong ability to equilibrium the whole search and local exploitation, and is easy to implement, with few design parameters, and has good applications in many fields. One study [15] improved AEO by introducing self-adaption nonlinear weight, optional cross-learning and Gaussian mutation strategies, as well as an excogitated IAEO algorithm to handle the optimization problem of PEMFC. In [16], the authors applied an EAEO to settle the actuality problem of distributed power distribution in the distribution system by combining AEO with the sine and cosine operator so as to cut down the power wastage of the distribution network. In [17], the authors proposed the MAEO algorithm to handle the parameter estimation of the PEMFC problem by introducing the linear operator H to balance exploration and exploitation. The authors of [18] proposed a promising blend multi-population algorithm HMPA for managing multiple engineering optimization problems. HMPA adopts a new population division method, which divides the population into several subpopulations, each of which dynamically exchanges solutions. An associative algorithm based on AEO and HHO was presented. Regarding this knowledge, multiple strategies were mixed to get the maximum efficiency of HMPA. For [19], the authors proposed a new random vector function chain (RVFL) network combined with the AEO algorithm to calculate the optimal outcome of the SWGH system. The authors of [20] proposed a new fitness function reconstruction technology based on the AEO algorithm for photovoltaic array reconstruction optimization. The authors of [21] used an AEO algorithm to acquire the ideal modulation parameters of a PID controller for an AVR system.

Although the AEO algorithm has been successfully used in the above domains, the AEO algorithm first explores all individuals and then executes the exploitation in sequence mode, which aggrandizes the computational burden and retards the optimization speed of the AEO. Secondly, the correcting of the AEO in the exploration stage puts individuals at the end of the biological chain, which does not account for the biological competition in nature—that is, every creature preys on other species and is itself the target of prey, so the solution accuracy is not high. 

For the sake of making the AEO characteristics better and ameliorating the accuracy of an optimization scheme, inspired by environmental stimulation and rival mechanism, when the population diversity in the environment is large, there are fewer similar organisms distributed in unit space, and the exploration ability is strong. In this case, the exploitation should be strengthened, and vice versa. Based on this, this paper proposes a combination of environmental stimuli and biological competition mechanism improved the artificial ecological optimization algorithm (SIAEO), improvements on AEO to do the following two aspects: (1) the environmental stimulation mechanism is introduced into the AEO algorithm, the external environment stimulation to achieve defined by the population diversity population interaction perform manipulations of consumption and, reduce the computational complexity of algorithm; (2) In the consumption stage, considering the biological competition, an individual is added to die due to predation, and the maximum cumulative success rate of each individual performing different tasks is used to guide the individual to choose a more suitable consumption renewal strategy, which redounds to the exploitation of the algorithm in the exploration stage. In the decomposition stage, two arbitrarily chosen individuals are introduced to define a potential exploitation and update formula, which can excellently assist the algorithm apart from the local optimal so as to build on proportionate the overall search and local optimization of the algorithm.

K-means [22,23] is an excellent clustering algorithm in a complex clustering problem. After determining the number of classes as well as clustering centers, the nearest neighbor principle is used to allocate samples to the categories determined by K clustering centers, and the in-class distance is minimized, and the inter-class distance is maximized by constantly updating the K-clustering centers. Among them, the determination of K cluster centers belongs to the high-dimensional optimization problem.

To demonstrate the effectiveness and functionality of the SIAEO algorithm, this experiment consists of two parts. Firstly, 40 benchmark functions composed of the CEC2017 and CEC2019 are designed to measure the performance of SIAEO to reply to complex high-dimension numerical optimization, and the results are paralleled with nine other excellent algorithms. The numerical optimization potential of the SIAEO is proved. Secondly, the validity of SIAEO solving the engineering optimization and K-means optimal clustering center was verified, and four engineering optimization problems and nine criterion data sets of a UCI machine learning knowledge base were clustered. The results confirmed the significance and competitiveness of the SIAEO–K-means algorithm.

The structure of this article is that Section 2 outlines the AEO, and Section 3 gives a list of the improvement strategy of the improved AEO algorithm. In Section 4, numerical examples and practical application examples and cluster optimization show the excellent performance of the SIAEO. In the conclusions, we review the entire paper and point out directions for future research.

## 2. AEO Algorithm

For the D-dimensional optimization problem:$$\min\;f(x_{1},x_{2},\ldots x_{D})\;s.t\;\;low_{j}\leq x_{j}\leq up_{j},j=1,2,\ldots ,D$$
where $f(x_{1},x_{2},\ldots x_{D})$ represents the optimization function, $x_{j}$ is the j-th decision variable and $up_{j}$ and $low_{j}$ are the upper and lower bounds of $x_{j}$, respectively.

The AEO algorithm simulates the energy flow of the ecosystem, which is presented in 2019. According to the composition of the food chain, it is divided into three stages: production, consumption and decomposition. The detailed steps for solving the optimization problem AEO algorithm are:
(1)Initialization: Suppose N expresses the population size, randomly generate N individuals in the D-dimensional decision space to compose the early generation population ed by $X(0)={\left\{X_{i}(0)\right\}}_{i=1}^{N}$, the *j*-th dimension of the individual $X_{i}(0)={\left\{x_{ij}(0)\right\}}_{j=1}^{D}$ is created by
(1)$$x_{ij}(0)=rand(0,1)\cdot (up_{j}-low_{j})+low_{j}\;.$$
in which rand(0,1) is a random number within (0, 1).(2)Production phase: Suppose the *t*-th population is $X(t)={\left\{X_{i}(t)\right\}}_{i=1}^{N}$, which are ranked in descending order with optimal value, in the production stage, only a temporary location update is performed on producer $X_{1}(t)$. The specific update strategy is a linear combination of best $X_{N}(t)$ and randomly generated individual $X_{r}(t)$, and the updated formula is:(2)$$X_{1}^{new}(t+1)=(1-a)\cdot X_{N}(t)+a\cdot X_{r}(t).$$
where $a=r_{1}\cdot (1-\frac{t}{T})$ expresses the coefficient used for linear weighting, $X_{r}(t)={\left\{x_{rj}(t)\right\}}_{j=1}^{D}$, $x_{rj}(t)=low_{j}+rand(0,1)\cdot (up_{j}-low_{j})$, represents a member arbitrarily generated in the solution space; t,T respectively represent this iteration and the final iteration. $r_{1}$ indicates a random value between [0, 1]. Equation (2) shows that in the pre-development stage of the algorithm, producers tend to explore $X_{r}(t)$ extensively, while in the later stage, focus on further exploitation near $X_{N}(t)$.(3)Consumption phase: In this stage, the second to *N*-th individuals are temporarily updated according to the random walk strategy generated by Levy’s flight. Let $\;v_{1}$ and $v_{2}$ be two random numbers with standard normal distribution and the consumption factor is defined:(3)$$C=\frac{1}{2}\frac{v_{1}}{\left|v_{2}\right|},\;\;v_{1}\sim N(0,1),\;\;v_{2}\sim N(0,1).$$
in which N(0,1) represents the normal distribution.The *i*-th individual was classified and updated according to probability by using the formula of herbivorous (Task 1), carnivorous (Task 2) and omnivorous (Task 3), i.e.,
(4)$$X_{i}^{new}(t+1)=\left\{ \begin{array}{l} X_{i}(t)+C\cdot (X_{i}(t)-X_{1}(t)),\;i\in [2,\ldots N],\begin{array}{cc} & 0<r_{2}\leq \frac{1}{3}\begin{array}{cc} & \;\;\;(\mathrm{Task}1) \end{array} \end{array}\\ X_{i}(t)+C\cdot (X_{i}(t)-X_{k}(t)),\;i\in [3,\ldots N],\begin{array}{cc} & \frac{1}{3}<r_{2}\leq \frac{2}{3}\begin{array}{cc} & \;\;\;\;(\mathrm{Task}2) \end{array} \end{array}\\ X_{i}(t)+C\cdot (r_{3}\cdot (X_{i}(t)-X_{1}(t))\\ \begin{array}{ccc} & & \begin{array}{cc} & +(1-r_{3})(X_{i}(t)-X_{k}(t))) \end{array} \end{array},\;i\in [3,\ldots N],\begin{array}{cc} & \frac{2}{3}<r_{2}<1\begin{array}{cc} & \;\;\;(\mathrm{Task}3) \end{array} \end{array} \end{array}\right..$$
where k is a positive integer with a random value in [2,i−1], $r_{2},r_{3}$ are the random values in [0, 1].(4)Decomposition phase: The temporarily updated population $X^{new}(t+1)$ is further revised to obtain the final revised population:(5)$$X_{i}(t+1)=X_{N}(t+1)+3u\cdot (e\cdot X_{N}(t+1)-h\cdot X_{i}^{new}(t+1)).$$
where u is a random number that is normally distributed, $e=r_{4}\cdot rand(1,2)-1$, $h=2\cdot r_{4}-1$ and $r_{4}$ is a random number between [0, 1].

In summary, the AEO algorithm wantonly creates initial solutions. In the process of updating and optimizing the population of each generation, the position of the producer is revised by relying on Equation (2), while the updating of consumers are carried out according to the isoprobability decision method by using Equation (4) and the final decomposition process is performed by Equation (5) until the termination conditions are met, the algorithm ends.

## 3. An Improved Interactive Artificial Ecological Optimization Algorithm Based on Environmental Stimulus and Biological Competition Mechanism

In an ecosystem, the survival and death of various organisms in the ecosystem, the predation of animals, and the stability and balance of the ecological chain are all greatly affected by the environment. In the process of predation, animals respond to different environmental stimuli in order to cope with the impact of the intensity of environmental stimulus on the predation patterns of animals. The incentive mechanism is a mechanism to describe the relationship between environmental stimulus and individual response. Combined with the AEO algorithm, environmental stimulus S is defined to guide individuals to carry out task conversion between consumption and decomposition, thereby meaning that consumption and decomposition can be executed interactively rather than sequentially, thereupon then reducing the complexity of the algorithm. In the process of predation, animals have incentive responses to different environmental stimuli and form their own preferences that are influenced by natural enemies. Furthermore, since the survival of organisms itself follows the competition mechanism of survival of the fittest, the introduction of competitive trend in the consumption stage can better simulate the real situation of the ecosystem. 

The purpose of this paper is to better improve the exploration and development ability of the AEO algorithm. Therefore, we introduced an interactive execution environment to stimulate the consumption and decomposition of tasks through the incentive mechanism and biological competition relations into the AEO algorithm. The consumption phase of the population individuals changes by searching the environment, and the largest accumulation of mission success rates guides the best-performing way to feed. The method of isoprobability execution of the AEO is abandoned, the convergence ability of the AEO is accelerated, and the exploitation function of the AEO in the exploration stage is intensified. At the same time, during the decomposition stage, through the introduction of arbitrary individuals, the exploration ability in the exploitation stage is boosted.

### 3.1. Environmental Stimulus

Environmental stimulus describes the external drive of an individual to perform a task. In the process of an algorithm search, the execution opportunity of consumption and decomposition can be measured by population diversity.

Population diversity is an indicator that describes the differences between individuals and reflects the distribution of populations [24]. For the *t*-generation population, population diversity [25] was measured as:(6)$$Diversity(t)=\frac{1}{N\cdot L}\sum_{i=1}^{N}\sqrt{\sum_{j=1}^{D}{\left(x_{ij}(t)-{\bar{x}}_{j}(t)\right)}^{2}}.$$
(7)$${\bar{x}}_{j}(t)=\frac{\sum_{i=1}^{N}x_{ij}(t)}{N}.$$
where N and L represent the size of population and reconciliation space, respectively, D is the dimension, $\bar{X}(t)$ is the population center, $\bar{X}(t)={\left\{{\bar{x}}_{j}(t)\right\}}_{j=1}^{D}$.

Environmental stimulus is defined as:(8)$$S(t)=\sqrt[{p}]{Diversity(t)}.$$
as well as p indicates the sensitivity coefficient of the stimulus to diversity. Referring to the value in reference [25] and simulation experiment, p is 50 in this paper.

When S(t)≥1, the variety of the group is better and the current algorithm is in the exploration stage. This moment increasing the exploitation potential of the algorithm while carrying out exploration can be helpful to improve the calculation accuracy. Therefore, when S(t)≥1, the consumption stage of AEO algorithm is implemented, and the updated formula is modified using a competition relationship to enhance the exploitation ability. When S(t)<1, the population diversity is relatively poor, and the individuals of the population have converged to the vicinity of the optimal solution. At this time, the exploitation should be strengthened to help the algorithm seek out the best solution, nevertheless at the same time, the scope of the population should be maintained to forestall individuals from getting bogged down in the local optimal. Therefore, when S(t)<1, the decomposition stage of AEO algorithm is implemented in which the individuals emulate from the best individuals as well as the random mean is integrated to enhance the exploration performance. To boost the global and local searching of the AEO, the calculation precision of the algorithm is improved.

### 3.2. Incentive Mechanism

In the incentive mechanism, the individual’s success in performing a search task refers to the improvement of its fitness, and the individual’s propensity to perform a task is measured by the cumulative success rate. The cumulative success rate should increase when the individual successfully performs the task and decrease when the individual fails to perform the task. The greater the cumulative success rate, the greater the propensity to perform the task, and vice versa.

By absorbing the advantages of the incentive mechanism, the three different consumer predation modes (Task 1), (Task 2) and (Task 3) in the consumption stage are regarded as three different search tasks. Therefore, we utilize the cumulative times of individual i successfully executing task j until the t-th iteration to calculate the cumulative success rate. The specific methods are as follows:

(a)Cumulative success rate initialization. For each individual $X_{i}(0)=(x_{i1}(0),x_{i2}(0),\cdots ,x_{iD}(0))$ in the initial population, a three-dimensional cumulative success rate vector $CSR_{i}(0)=(CSR_{i1}(0),CSR_{i2}(0),CSR_{i3}(0))$ is initialized, where $CSR_{ij}(0)\in [0,1]$ represents the initial cumulative success rate of the i-th individual performing the j-th task. The assignment method is:(9)$$CSR_{i1}(0)=CSR_{i2}(0)=rand(0,0.4),$$
(10)$$CSR_{i3}(0)=1-CSR_{i1}(0)-CSR_{i2}(0).$$
where rand(a,b) stands for the random value in [a,b], and Formula (10) ensures that the sum of the elements after initialization is 1. Then initialize the three dimensional count vector $Count_{i}(0)=(Count_{i1}(0),Count_{i2}(0),Count_{i3}(0))$ to store the times of individual i successfully executing task j, where $Count_{ij}(0)=0$.(b)Cumulative success rate update process. Firstly, the update process of count vector $Count_{i}(t)$ is introduced. Let $Flag_{ij}(t)$ represent the identifier of the j-th task performed by the i-th individual of in the t-th generation. When individual *i* performs task *j*, let $Flag_{ij}(t)=1$; otherwise, it is 0. Since each individual in each generation performs only one of the three tasks in Equation (4). Therefore, for individual *i*, only one element of $Flag_{ij}(t)$ (j=1,2,3) is 1. Let $X_{i}^{new}(t)$ represent the new individual obtained by individual *i* of in the *t*-th generation after performing task *j,* then the $Flag_{ij}(t)=1$. If $X_{i}^{new}(t)$ is superior to $X_{i}(t)$, that is, individual *i* successfully performs task *j*, then the possibility of executing task *j* next time should be increased while the chance of executing other tasks should be weakened. The j-th element of the corresponding count vector $Count_{i}(t+1)$ increases by 1 while the remaining elements remain unchanged, that is,
(11)$$\{\begin{array}{c} Count_{ij}(t+1)=Count_{ij}(t)+1\\ Count_{ik(k\neq j)}(t+1)=Count_{ik((k\neq j))}(t) \end{array}\begin{array}{cc} & Flag_{ij}(t)=1\;\&\&\;fit(X_{{}^{i}}^{new}(t))<fit(X_{i}(t)) \end{array}$$
where fit(•) refers to the individual’s fitness, which is defined as the objective function value in this paper.

If $X_{i}^{new}(t)$ is inferior to $X_{i}(t)$, that is, individual *i* fails to execute task *j*, then the possibility of executing task *j* next time should be reduced and the chance of executing other tasks should be increased. The j-th element of the corresponding count vector $Count_{i}(t+1)$ remains unchanged, while the current value of other elements increases by 1, that is
(12)$$\{\begin{array}{c} Count_{ij}(t+1)=Count_{ij}(t)\\ Count_{ik(k\neq j)}(t+1)=Count_{ik(k\neq j)}(t)+1 \end{array}\begin{array}{cc} & Flag_{ij}(t)=1\;\&\&fit(X_{{}^{i}}^{new}(t))\geq fit(X_{i}(t)) \end{array}$$

At this point, the vector $CSR_{i}(t+1)$ of the cumulative success rate of the next generation of individual *i* is updated with the count vector $Count_{i}(t+1)$ as:(13)$$CSR_{ij}(t+1)=\frac{Count_{ij}(t+1)}{2\cdot (N-1)\cdot t}.$$

For each individual, cumulative success rate vector $CSR_{i}(t+1)=(CSR_{i1}(t+1),\;CSR_{i2}(t+1),\;CSR_{i3}(t+1))$ can be calculated. Let $CSR_{i*}(t+1)=\underset{1\leq j\leq 3}{\max}\{CSR_{ij}(t+1)\}$ and take $CSR_{i*}(t+1)$ as a definite indicator of the task to be executed in the next consumption phase. In the consumption phase of the AEO, the corresponding updatings is executed with equal probability without considering the incentive mechanism. Now, formula (13) is used to calculate the cumulative success rate of each individual performing the three tasks, and the strategy corresponding to the * task is determined according to the indicator of task propensity $CSR_{i*}(t+1)$, so as to motivate individuals to explore more promising areas in a more appropriate way. The incentive mechanism pseudocode is named Algorithm 1.
**Algorithm 1:** Incentive mechanism (*N*, X(t), CSR(t), Flag)1For *t*-th generation population $X(t)={\left\{X_{i}(t)\right\}}_{i=1}^{N}$ 2Calculated population diversity Diversity(t) and environmental stimulus S(t) 3
 if S(t)≥1
4   for *i* = 1:*N*5
 $\;CSR_{i*}(t)=\max(CSR_{ij}(t))$
6Perform Task*; $Flag_{i*}(t)=1$ 7   end for  % Get the updated population $X^{new}(t)={\left(X_{i}^{new}(t)\right)}_{i=1}^{N}$ 8
 for i=1:N
9
       for j=1:3
10
  $if\;Flag_{ij}==1\&\&fit(X_{i}^{new}(t))<fit(X_{i}(t))$
11Update optimal solution and fitness value12$\;Count_{ij}(t+1)=Count_{ij}(t)+1,\;\;Count_{ik(k\neq j)}(t+1)=Count_{ik((k\neq j))}(t)\;else\;ifFlag_{ij}==1\&\&fit(X_{i}^{new}(t))>=fit(X_{i}(t))$13
       $Count_{ij}(t+1)=Count_{ij}(t),\;Count_{ik(k\neq j)}(t+1)=Count_{ik(k\neq j)}(t)+1$
14   end if15
    $CSR_{ij}(t+1)=Count_{ij}(t+1)/(2\cdot (N-1)\cdot t)$
16
 $Flag_{ij}(t+1)=0$
17
 end for
18
 end for
19
 else 
20   for *i*=1:*N*21Execute the decomposition phase22   end for23
 end if
Output: t+1 generation population

### 3.3. Updating Based on Competition Mechanism

Due to ecological competition, the individuals in the consumption stage of the ecosystem eat both the low-ranking organisms and the high-ranking natural enemies. In Equation (4), the influence of natural enemies is not taken into account, which makes the algorithm have strong exploration ability but poor exploitation ability in the consumption stage. In Equation (5), the algorithm is apt to stumble into local optimization as a result of only considering the lead of the optimal individual. When choosing the execution strategy of the algorithm according to the environmental stimulus, each strategy should have both exploration and exploitation performance. Therefore, the influence of predator predation is integrated into (4), and the Task to be executed is determined according to the maximum cumulative success rate. That is, if the $\;CSR_{i*}(t)=\max(CSR_{ij}(t))$, then the Task* is executed and updated as
(14)$$X_{i}(t+1)=\left\{ \begin{array}{l} X_{i}(t)+C\cdot (X_{i}(t)-X_{1}(t)),i\in [2,\ldots N].\begin{array}{cc} & \mathrm{Task}1 \end{array}\\ X_{i}(t)+C\cdot (r_{5}\cdot (X_{i}(t)-X_{k}(t))+(1-r_{5})(X_{j}(t)-X_{i}(t))),\;i\in [3,\ldots N].\begin{array}{cc} & \;\mathrm{Task}2 \end{array}\\ X_{i}(t)+C\cdot (r_{6}\cdot (X_{i}(t)-X_{1}(t))+(1-r_{6})(X_{i}(t)-X_{k}(t))\\ \begin{array}{ccc} & & \begin{array}{cc} +r_{7}\cdot (X_{j}(t)-X_{i}(t))) & \end{array} \end{array},\;i\in [3,\ldots N].\begin{array}{cc} & \mathrm{Task}3 \end{array} \end{array}\right.$$
where k means a positive integer of the random value in interval [2,i−1], j indicates a positive integer of the random value of [i+1,N], and $r_{5},r_{6},r_{7}$ are random numbers at (0, 1).

In Equation (5), the drag of random individuals is integrated into enabling the algorithm to from local optimum. For two randomly selected individuals $X_{rand1}(t)$ and $X_{rand2}(t)$ in the current population, a more reasonable updating formula is put forward to substitute for the decomposition strategy of the original algorithm. Formula (15) is adopted to displace the intrinsic individual position updating strategy of Formula (5):(15)$$X_{i}(t+1)=X_{N}(t+1)+3u\cdot (e\cdot X_{N}(t+1)-h\cdot X_{i}^{new}(t+1))+\frac{X_{rand1}(t)+X_{rand2}(t)}{2}.$$

Other parameters are the same as Equation (5). In Equation (15), the insertion of two heterogeneous individuals in the population can perfectly heighten the universality of exploration of the algorithm and aid the algorithm in averting the local optimum.

### 3.4. Implementation Steps of SIAEO Algorithm

To make the above preparations, the incentive mechanism and competition relationship were integrated into AEO, the environmental stimuli were defined according to population diversity, and the next-generation search strategy was determined according to the cumulative success rate of the search tasks, so as to effectively make the convergence speed and the precision in the calculation of the AEO better. The global searching and local optimization of the AEO balance are further promoted by modifying the updated formula of consumption and decomposition stage through competition relationship. Based on this, this paper provides an improved artificial ecological optimization algorithm (SIAEO) that integrates into environmental stimulus and competition mechanism. The execution programs of the SIAEO are described as:

Step 1: Set algorithm parameters.

Step 2: According to Equation (1), the initial population $X(0)={\left\{X_{i}(0)\right\}}_{i=1}^{N}$ is randomly generated and the cumulative success rate of each individual performing three different search tasks is initialized and assigned.

Step 3: The diversity (Diversity) of the population and the environmental stimulus S were calculated through Equations (6)–(8), and the individuals were rearranged in ascending order in light of the fitness value.

Step 4: According to the numerical value of S, the corresponding strategy is executed to obtain the new population, as follows:

If S≥1, execute the consumption operator, each individual adopts the corresponding formula in Formula (14) to update according to the maximum cumulative success rate; The cumulative success rate of each individual performing three search tasks was calculated;

If S<1, then use Equation (15) to perform the decomposition operator.

Step 5: Judge whether t<T is true; if so, make t=t+1, then the algorithm turn to Step 3. Otherwise, the algorithm ends and outputs the best solution and corresponding function value.

The working frame of the SIAEO is displayed in Figure 1.

### 3.5. Time Complexity Analysis of the Algorithm

In the SIAEO, if the meanings of N, D and T are shown above, then the time complexity of the beginning phase is O(N⋅D). The time complexity of the population diversity and the production are O(N⋅D⋅T) and O(D⋅T), while consumption and the decomposition phase are both O(N⋅D⋅T). According to the incentive mechanism, only one of the two can be executed in an iteration, but because the consumption operator and the decomposition operator contain exploration and exploitation capacity in the meantime, the stimulus incentive mechanism properly uses these two operators and better balances the relationship between them. The complexity of evaluating fitness values for each individual is O(N⋅T). The complexity of sorting fitness values is O(N⋅logN⋅T). As a consequence, the total computational time complexity of the SIAEO algorithm is:(16)O(SIAEO)=O((N+T)⋅D+(2N⋅D+N+N⋅logN)⋅T)

## 4. Numerical Experiment

For the sake of testing the superiority of the SIAEO in settling complicated problems and the practical applicability of solving the actual problems, the latest standard test set CEC2019 was selected to separately analyze the validity of the improved strategy in Section 4.1. In Section 4.2, the CEC2017 test set was selected for high dimensional numerical experiments, and CEC2019 was selected to test the performance divergence between SIAEO and other algorithms. In Section 4.3, the SIAEO is applied to manage four engineering minimization problems. Finally, the high-dimensional K-means clustering problem is selected to declare the precision and accuracy of clustering by the SIAEO.

### 4.1. Strategy Effectiveness Analysis

To validate the influence of a single improvement strategy on the SIAEO algorithm, experiments were set up to compare the SIAEO algorithm with the SAEO that only appended environmental stimulus strategy, the CAEO that only added incentive mechanism and biological competition mechanism strategy in the composition stage, the DAEO that only increased local escape operator mechanism strategy in decomposition stage and AEO [2].

#### 4.1.1. Test Functions

The effectiveness of the improved strategies was evaluated using ten multi-modal functions of the CEC2019 standard test set [26]. Table 1 gives the range of variables, dimensions, and the theoretical optimal value $F(x^{*})$ of the CEC2019 test function. Although the dimension of the CEC2019 test set is not high, each function has many local optimal advantages, among which f4, f6, f7 and f8 have many local optimal advantages, which tremendously challenges the global minimization of the intelligence algorithm.

#### 4.1.2. Strategy Effectiveness Verification of the SIAEO

Table 2 compares the outcomes of the SIAEO with the SAEO, CAEO, DAEO and AEO algorithms on CEC2019 test set. The same parameter settings were used for all five algorithms, namely, an independent running for 30 times, N=50, the estimation times of the maximization function is N⋅10000 times, average Ave and standard deviation Std represent the error between the optimal value and the theoretical value of 30 times running and average CPU Time of 30 times running was calculated. Two-tailed *t*- and Friedman tests were applied to inspect the statistical comparison of all the methods of concern. Where the bold data represents the optimal results calculated by the five algorithms, (+), (=) and (−) respectively signify the use of a two-tailed *t*-test at a significance level α=0.05, the SIAEO is superior to, equal to and inferior to the comparison algorithm, and (#) line remarks the number of corresponding results. The Friedman rank means the sorting result of the Friedman test.

It can be noted from Table 2 that, for the 10 functions of the CEC2019, the solution results of the five algorithms for the f1 function have reached the optimal value. The SAEO, CAEO, DAEO, and SIAEO each have advantages over the other test functions. Among them, the SAEO has the best performance in solving the f3 and f4 functions. The CAEO possessed the optimal mean and standard deviation in the f2 function. While the DAEO acquired the best average results on three functions (f6, f8 and f9), the SIAEO captured the best average results on the f5, f7 and f10 functions. On the whole, the addition of each strategy individually contributes to the advancement of the AEO algorithm in some way. For different test functions, each strategy has distinct effects. Therefore, the SIAEO algorithm gathered in one place with the three improved strategies is more promising and competitive.

#### 4.1.3. Analysis of Statistical Test Results

The last four lines of Table 2 clearly state that the results of two statistical tests, the two-tailed *t*-test and the Friedman test, demonstrate the absolute superiority of our improved SIAEO algorithm. The number of functions similar to the SIAEO, SAEO, CAEO and DAEO algorithms is 2, 6, 4 and 7, respectively, while the number of functions inferior to SIAEO is 7, 3, 5 and 2, respectively. On the one hand, this manifests that the SIAEO algorithm is provided with significant advantages and superior performance over the AEO algorithm and shows the effectiveness of the improvement strategy at the same time. On the other hand, we can also discover the importance of the orientation and influence degree of the three strategies in the iterative optimization process. The DAEO algorithm, which only raises the local escape operator in the decomposition stage, gains more proficiency than other strategies. As many as seven functions are close to the SIAEO’s performance. The SAEO, which only attaches environmental stimulus mechanisms to reduce computational complexity, obtains six functions close to the SIAEO’s performance.

The Friedman test showed an ability to test the similarity of the five algorithms, with smaller results indicating better algorithms. The SIAEO algorithm took the minimum value of 2.419 and ranked first overall, followed by the DAEO, CAEO, SAEO and AEO algorithms to demonstrate the superiority of the improved strategy.

#### 4.1.4. The Tendency Analysis of Consumption Operator

There are three predation modes in the consumption stage, showed by Task1, Task2 and Task3. The algorithm changes from overall situation performance to local performance with iterations increasing. However, the original algorithm adopts three tasks with equal probability execution, which will cause a part of resource waste in the late iteration. The incentive mechanism will improve this situation to a large extent. The number of tasks performed by Task2 is much higher than that of other tasks, followed by Task3, and the number of tasks performed by Task1 decreases in the final stage of the iteration. In each iteration, we recorded the frequency of individuals performing the three tasks and stacked them up as we went through the iteration. Figure 2 shows the running results on the node test functions of the CEC2019.

In Figure 2, it can be seen that in the prophase iterations of the SIAEO, the algorithm pays attention to global exploration ability and executes the consumption operator. The cumulative number of tasks performed increases with each iteration. The improved carnivorous and weedy strategies have a higher probability of being selected because they combine exploration and exploitation, resulting in a higher slope of the cumulative line. At the end of the iteration, environmental stimuli guide the algorithm to perform decomposition operations. Due to the termination of the consumption operator, the individual does not perform the three tasks, so the cumulative curve presents a horizontal straight line in the late iteration.

### 4.2. Numerical Optimization Experiment

#### 4.2.1. Parameter Setting

To show the availability and stability of the SIAEO in handling high-dimensional optimization problems, a total of 40 benchmark test functions of the CEC2019 [26] and CEC2017 [27] were numerically optimized. The CEC2017 test set is composed of a total of 30 functions, including single-peak functions (F1–F3), simple multi-peak functions (F4–F10), mixed functions (F11–F20) and composite functions (F21–F30). In the experiment, the SIAEO algorithm was compared with the AEO’s improved algorithms IAEO [15], EAEO [16] and AEO [14], as well as other representative swarm intelligence algorithms, including AOA [28], TSA [29], HHO [2], QPSO [30], WOA [31] and GWO [32]. Table 3 shows the parameter values of each algorithm used in the experiment. To be fair, all algorithms run independently 30 times, N=50 and, D=100 and the maximum estimation times is N⋅10000 times. The computer is configured as Intel(R) Core I7, main frequency 3.60 ghz, memory 8 GB, Windows 7 64-bit operating system, and the programming is by MATLAB R2014a.

#### 4.2.2. Comparison of Results between SIAEO and Comparison Algorithm

Table 4 and Table 5 exhibit that the SIAEO and nine other algorithms run the CEC2017 and CEC2019 test sets independently 30 times. The Ave and Std and the average CPU Time of 30 times are calculated. The bold data display the minimum value obtained by each algorithm. The symbols “+”, “−” and “=“ in () after time in the table indicate that the SIAEO is significantly better than, worse than or similar to the competitive algorithm when the *t*-test is executed on the SIAEO and the other nine comparison algorithms. The last four rows of Table 4 and Table 5 show the number of best solutions achieved on all test functions ((#) best), as well as the *t*-test ‘+’, ‘−’, and ‘=’ quantity statistics and Friedman test results.

The data in Table 4 demonstrate that the SIAEO performed best for the high-dimensional and challenging CEC2017 test set in line with the number of optimal solutions achieved ((# best) with a value of (# best) of 21. Specifically, the SIAEO accessed the best results on six of the seven simple multi-peak functions (F4–F10). For mixed functions (F11–F20) and composite functions (F21–F30), the performance is best on seven and eight functions, respectively. For three unimodal functions (F1–F3), the performance is slightly inferior to that of the EAEO. In general, the SIAEO performs slightly less well for unimodal functions, but its performance on simple multimodal functions, mixed functions, and composite functions is the the most promising. Thus, the SIAEO algorithm possesses more advantages and more comprehensive performance in solving high-dimensional complex optimization problems.

The data in Table 5 illustrate that the SIAEO’s (#)best value is nine in terms of the number of optimal solutions ((#)best), which indicate that the SIAEO unfolds before one’s eyes a significant advantage in solving the CEC2019 test set with fixed dimensions. In conclusion, the SIAEO algorithm shows more promising performance for both the high-dimensional CEC2017 test set and the fixed-dimensional CEC2019 test set, which authenticates that the SIAEO outperforms the other nine comparison algorithms in terms of scalability.

Figure 3 provides the comparison diagram of curve convergence process of three simple multi-peak functions (F5, F6, and F8), three mixed functions (F15, F16, and F19), and three composite functions (F21, F23, and F24) of the CEC2017 test set when the SIAEO and other algorithms search for optimization. For ease of observation, the convergence curve is plotted with log10(F(x)) as the ordinate. Figure 3 illustrates this point: although the SIAEO algorithm is slower than AEO algorithm in solving simple multi-peak, mixed function and composite functions, it has strong exploration and exploitation performance. It is obviously better than the other nine reference algorithms in terms of solving precision and has a strong ability to escape the local optimal.

#### 4.2.3. Analysis of Statistical Test Results

Statistical test [33] is an essential mathematical method to analyze the results of intelligent optimization algorithms. When the significance level is α=0.05, three statistical tests, including the two-tailed *t*-test, Wilcoxon signed-rank test and Friedman test, are used to test the performance of the comparison algorithm.

(1)Two-tailed *t*-test

In Table 4 and Table 5, the *t*-test results of the SIAEO and the other nine comparison algorithms are given in parentheses after Time, and (#) best gives the number of winning results. As an attempt to comprehensively assess all algorithms, the comprehensive performance CP is defined by the number of superior algorithms minus the number of inferior algorithms, i.e., “(#)+” − “(#)−”. When the CP is positive, that is, CP > 0, it indicates that the SIAEO is superior to the comparison algorithm and vice versa. Table 6 shows CP values of the SIAEO and other comparison algorithms on the CEC2017 test set and the CEC2019 test set. Obviously, the data in Table 6 indicates that the SIAEO is an excellent algorithm for the CEC2017 test set. The CP value of the SIAEO compared with the AEO is 13, which exhibits the superiority of the SIAEO amendment scheme. For the EAEO and IAEO, the CP values are 29 and 16, respectively, which suggests that the proposed algorithm is significantly superior to the EAEO and IAEO. By comparing the CP values of the other six intelligent algorithms, the SIAEO achieved an overwhelming superiority, with the CP values of thirty compared with the other four algorithms, except that the CP values of WOA and HHO algorithms are 28. For the CEC2019 test set, the CP values of the SIAEO algorithm are positive compared with the AEO, EAEO and IAEO, while the CP values of the other six comparison algorithms are close to the maximum. In all, it shows that the SIAEO algorithm has more advantages and prospects in exploring high-dimensional minimize problems.

(2)Wilcoxon signed-rank test

For the CEC2017 and CEC2019 test sets, the Wilcoxon signed-rank test with a significance level of 0.05 was used to test the difference between the SIAEO and other competitive algorithms. In Table 7, R+ and R− represent the sum of the rank of all the test functions in which the SIAEO is better or worse than its competitors. The greater the difference between R+ and R-, the better the performance of the algorithm will be. *p* represents the probability value corresponding to the test result. When *p* < 0.05, it means that a significant difference exists in the two algorithms. Significance means the significance of the difference, “+” expresses that the underlying algorithm is prominently better than the comparison algorithm, and “=” indicates that the performance is similar to the comparison algorithm. Table 7 shows that the SIAEO significantly outperforms the other nine algorithms except for the IAEO for the CEC2019 test set. Not only is *p* < 0.05, but the number of R+ is far greater than the number of R-, especially for the EAEO, AOA, TSA, HHO, QPSO, WOA and GWO, the number of R− is 0. For the CEC2017 test set, although the SIAEO is not significantly superior to the EAEO and AEO in *p*, it provides more potential performance than the EAEO and AEO algorithms in R+ and R−. In general, the SIAEO has a significant advantage among the nine comparison algorithms based on the *p*-value. Moreover, it outperforms all competitors in the number of R+ and R- and shows a more promising performance in high-dimensional cases.

(3)The Friedman test

Nonparametric Friedman test [34] was used to test the overall performance of the ten comparison algorithms. During the inspection, the average value of 30 times of independent operation is used as input data to calculate the overall ranking of each algorithm. Friedman test results and rankings for the CEC2017 and CEC2019 test sets are shown in the last row of Table 4 and Table 5. The results in Table 4 and Table 5 indicate that the SIAEO ranked first in both the 100 dimensions of the CEC2017 and the fixed dimensions of the CEC2019. The AEO, EAEO and IAEO rank 2nd, 3rd and 4th, respectively. In addition, the ranking does not change with the change of dimension, so illustrating the SIAEO has significant robustness and the ability to solve high-dimensional problems. Figure 4 shows the differences in Friedman ranking and illustrates the SIAEO having prominent preponderance over the other algorithms.

### 4.3. Engineering Optimization Experiment

In this subsection, to validate the ability of the SIAEO to solve the optimization problems in real-world applications, four real-world engineering challenges are disposed of availing on the proposed algorithm. In the comparison results, the optimal value is represented in bold font.

#### 4.3.1. Three-Bar Truss Design Problem

This example discusses a 3-bar planar truss structure (Figure 5) that minimizes volume while meeting stress restrictions on each side of the truss members [35]. Optimization of two-stage rod length $A_{1},A_{2}$ with the variable vector $x=(x_{1},x_{2})=(A_{1},A_{2})$ can be mathematically simulated as below:$$\begin{array}{l} \min f_{eo1}(x)=(2\sqrt{2}x_{1}+x_{2})l\\ s.t.\\ g_{1}(x)=\frac{\sqrt{2}x_{1}+x_{2}}{\sqrt{2}x_{1}^{2}+2x_{1}x_{2}}P-\sigma \leq 0\\ g_{2}(x)=\frac{x_{2}}{\sqrt{2}x_{1}^{2}+2x_{1}x_{2}}P-\sigma \leq 0\\ g_{3}(x)=\frac{1}{x_{1}+\sqrt{2}x_{2}}P-\sigma \leq 0\\ 0\leq x_{1},x_{2}\leq 1. \end{array}$$

The findings of the SIAEO and other excellent comparative methodologies are given in Table 8. Table 8 also lists the best decision variables of the optimal solution for all comparing approaches. Table 8 shows that by supplying optimal variables at $x^{*}=(x_{1}^{*},x_{2}^{*})=(0.788675136,0.40824828)$ with minimum objective function value: $f_{eo1}(x^{*})=263.895843$, the proposed SIAEO achieved better and comparable outcomes than other potential minimized methods.

#### 4.3.2. Himmelblau’s Nonlinear Problems

The mathematical description of Himmelblau’s nonlinear problems [40] using the vector $x=(x_{1},x_{2},x_{3},x_{4},x_{5})$ can be surveyed as:$$\begin{array}{l} \min f_{eo2}(x)=5.3578547x_{3}^{2}+0.8356891x_{2}x_{5}+37.293239x_{1}-40792.141\\ s.t.\\ 0\leq g_{1}(x)=85.334407+0.0056858x_{2}x_{5}+0.0006262x_{1}x_{4}-0.0022053x_{3}x_{5}\leq 92,\\ 90\leq g_{2}(x)=80.51249+0.0071317x_{2}x_{5}+0.0029955x_{1}x_{2}-0.0021813x_{3}^{2}\leq 110,\\ 20\leq g_{3}(x)=9.300961+0.0047026x_{3}x_{5}+0.0012547x_{1}x_{3}+0.0019085x_{3}x_{4}\leq 25,\\ 78\leq x_{1}\leq 102,33\leq x_{2}\leq 45,27\leq x_{3},x_{4},x_{5}\leq 45. \end{array}$$

The best results obtained using various methods are demonstrated in Table 9, which reveal that the SIAEO outperforms existing approaches and also has engineering practicability.

#### 4.3.3. Tabular Column Design Problem

Figure 6 illustrates the uniform tabular column design problem [44]. Generating a homogeneous column of tabular design at the lowest possible cost is its purpose. The following is a mathematical description of the problem using the variable vector $x=(x_{1},x_{2})$.
$$\begin{array}{l} \min f_{eo3}(x)=9.82x_{1}x_{2}+2x_{1}\\ s.t.\\ g_{1}(x)=\frac{P}{\pi \sigma_{y}x_{1}x_{2}}-1\leq 0,\\ g_{2}(x)=\frac{8PL^{2}}{\pi^{3}Ex_{1}x_{2}(x_{1}^{2}+x_{1}^{2})}-1\leq 0,\\ g_{3}(x)=2.0/x_{1}-1\leq 0,\\ g_{4}(x)=x_{1}/14-1\leq 0,\\ g_{5}(x)=0.2/x_{2}-1\leq 0,\\ g_{6}(x)=x_{2}/0.8-1\leq 0,\\ 2\leq x_{1}\leq 14,0.2\leq x_{2}\leq 0.8. \end{array}$$

Although the optimization variables of the tabular column design problem are few, there are many constraints that increase the difficulty of hunting for feasible solutions. The best results of each algorithm for searching such a problem are given in Table 10. The SIAEO method’s variation is clearly lower than the other approaches; in addition, the SIAEO method’s best result has the best performance by supplying optimal variables at $x^{*}=(x_{1}^{*},x_{2}^{*})=(5.4512,0.29167)$ with a minimum objective function value: $f_{eo3}(x^{*})=26.526$.

#### 4.3.4. Gas Transmission Compressor Design Problem

This four-variable mechanical design problem was first proposed by Beightlerand Phillips designed the problem [46]. The goal is to seek out the variable for sending natural gas to the gas pipeline transmission system at the lowest possible cost. The following is a mathematical description of the problem with the vector $x=(x_{1},x_{2},x_{3},x_{4})$.
$$\begin{array}{l} \min f_{eo4}(x)=8.61\times 10^{5}x_{1}^{1/2}x_{2}x_{3}^{-2/3}x_{4}^{-1/2}+3.69\times 10^{4}x_{3}+7.72\times 10^{8}x_{1}^{-1}x_{2}^{0.219}-765.43\times 10^{6}x_{1}^{-1}\\ s.t.\\ g_{1}(x)=x_{4}x_{2}^{-2}+x_{2}^{-2}-1\leq 0,\\ 20\leq x_{1}\leq 50,1\leq x_{2}\leq 10,20\leq x_{3}\leq 50,0.1\leq x_{4}\leq 60. \end{array}$$

The objective function of this problem is complex and highly nonlinear, which puts forward higher requirements for the optimization algorithm.

The optimal results of different methods for solving such a problem are given in Table 11. The SIAEO method’s variation is clearly lower than the other approaches; in addition, the SIAEO method’s best result has the best performance by supplying optimal variables at $x^{*}=(x_{1}^{*},x_{2}^{*},x_{3}^{*},x_{4}^{*})=(50,1.17828,24.59259,0.38835)$ with a minimum objective function value: $f_{eo4}(x^{*})=2964895$.

### 4.4. Application of SIAEO in Clustering Problem

K-means clustering is a very effective clustering method [51,52]. When the number of classes is determined, the determination of the clustering center is the key to this method. The finding of the clustering center can be turned into the optimization problem of minimum class spacing. For the data set $Data=\{o_{1},o_{2},\ldots ,o_{V}\}$ composed of *V* samples, there are M classes in total, the cluster center of class i (i=1,2,⋯M) is $P_{i}$, and the number of data in the cluster of class *i* is $V_{i}$, then the clustering center $P_{i}$(i=1,2,⋯M) of K-means clustering is the optimal of the following optimization model.
(17)$$\min\sum_{i=1}^{M}\sum_{j=1}^{V_{i}}{\left\|o_{j}-P_{i}\right\|}_{2}^{2}$$

Attempting to demonstrate the SIAEO algorithm’s competitiveness in solving high-dimensional practical problems by verifying its optimization ability for high-dimensional clustering, this section uses the SIAEO to search for the optimal clustering center of K-means clustering.

#### 4.4.1. SIAEO—K-Means Algorithm

Table 12 gives the number of data instances, class-number Classes, Dimension Features of data and the number of dimensions of the decision variables in optimization problem (17) in nine data sets from the UCI standard database [53]. The process of solving optimization problem (17) with the SIAEO algorithm is as follows: Firstly, *N* Dimension individuals are generated as *N* groups of clustering centers. For each individual, that is, each group of clustering centers, the K-means clustering method is adopted to determine the number of data contained in each category. Individual fitness values are calculated using Formula (17) and determine the optimal individuals. Then, the SIAEO algorithm is used to update each individual until the optimal cluster center is found.

#### 4.4.2. Experimental Setup and Performance Evaluation

In the experiment, N=20 and T=100, and the independent operation was performed 30 times. The IAEO, EAEO, AEO, DE and PSO were selected to combine with K-means as the comparison algorithm. Table 13 provides the mean Ave, standard deviation Std and Rank gained by the SIAEO and comparison algorithm to search the optimization model established by Formula (17) of the data in Table 12. The last three lines of Table 13 give the evaluation index. The count indicates the number of data sets of the optimal result obtained by each optimization algorithm. The Avg_Rank represents the average ranking and Total Rank quantitatively expresses the contrast in the optimization ability of each algorithm. For nine data sets, Figure 7 depicts the search process of each contrast algorithm to reflect the characteristics of the algorithm and to explore the global optimal solution during operation.

#### 4.4.3. Comparison and Analysis of Results

Table 13 shows that when compared with other algorithms, the SIAEO acquires the ideal intra-class distance in seven data sets and ranks 1.33 on average, placing it first overall. The EAEO and IAEO came in second and third place, while the PSO and DE came in 4th and 5th place, respectively, while the AEO ranked last, which differed greatly from the SIAEO. Figure 7 shows that the SIAEO possesses a strong capability to escape the local minimum and explores the global minimum compared with other algorithms, making it an outstanding heuristic algorithm for solving high-dimensional clustering problems.

## 5. Conclusions

In artificial ecosystem optimization algorithms, the relationship between exploration and exploitation has always been the focus of research. In this paper, by introducing the environmental stimulus incentives mechanism, which was defined by the population diversity of the external environment stimulus to assist populations to realize the conversion between consumption and decomposition, the largest cumulative success rate by individuals performing different tasks to guide the consumption of individual choice was found to be more suitable for an update strategy. This method decreases the complexity of the AEO and improves the calculation precision.

At the same time, in the consumption stage, a new exploration and updating method that uses biological competition to make it have successfully random search features is proposed. Because of this, an improved artificial ecosystem optimization algorithm (SIAEO) on account of environmental stimulus incentive mechanisms and biological competition was proposed. Two types of tests were developed to confirm the superiority of the SIAEO. The validity of the SIAEO and other intelligent algorithms and the AEO variant algorithms to resolve the CEC2017 and CEC2019 benchmark functions was confirmed in the first set of experiments. The SIAEO has a greater solving precision and convergence speed than contrast algorithms, according to the findings of the experiments. The SIAEO’s better stability and robustness are further demonstrated through statistical tests and the convergence curve.

The second group of experiments verified the usefulness of the SIAEO. Four engineering minimize problems verify the efficiency of the SIAEO in light of the complex engineering nonlinear search problems. The SIAEO–K-means model was established to optimize the K-means clustering center, and nine UCI standard data sets were applied in the experiment. The results showed that SIAEO–K-means obtained higher evaluation index values and had better performance in high-dimensional clustering data sets. The SIAEO exceeds the comparative algorithms according to optimization ability and performance in addressing high-dimensional problems in light of the two groups of experimental data.

As the AEO is a new heuristic algorithm, a more effective improvement strategy to balance its exploration and exploitation ability is studied to improve its optimization ability. The authors of [54] provided a dynamic data flow clustering method based on intelligent algorithms, providing a reference for the fusion of dynamic data clustering and heuristic algorithms. Another study [55] required an optimized estimation of hydraulic jump roller length. These new requirements need further exploration of the large-scale clustering ability of AEO algorithm and its deep application in the mechanical field in the future.

## Figures and Tables

**Figure 1 biomimetics-08-00242-f001:**
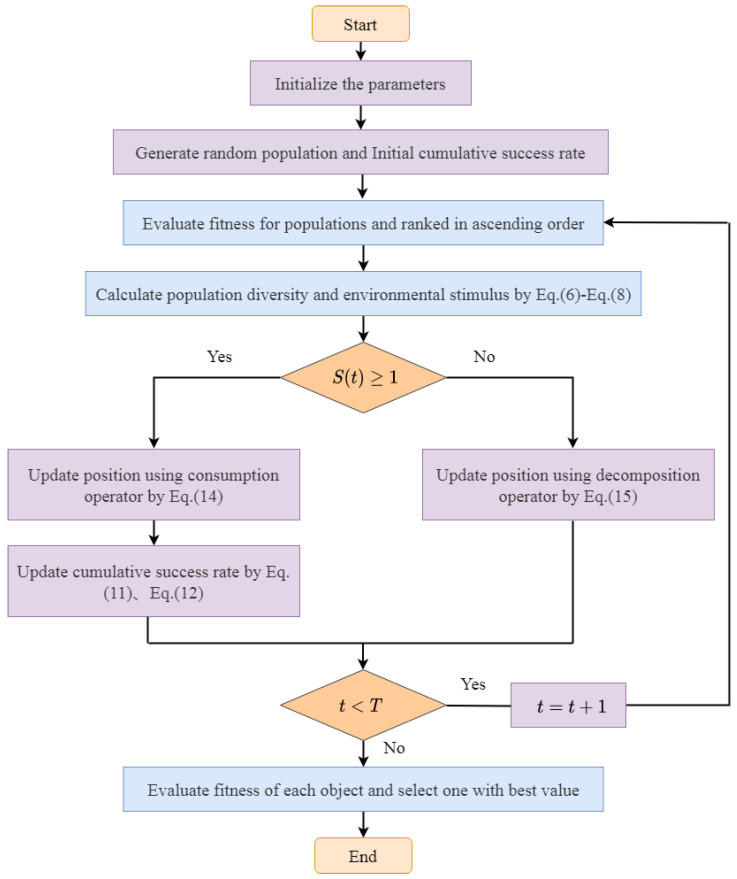
The flowchart of the SIAEO algorithm.

**Figure 2 biomimetics-08-00242-f002:**
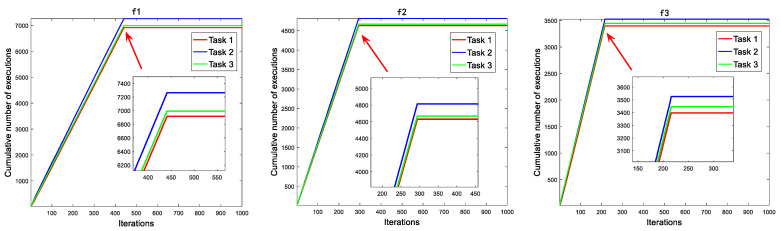
The cumulative number of tasks performed.

**Figure 3 biomimetics-08-00242-f003:**
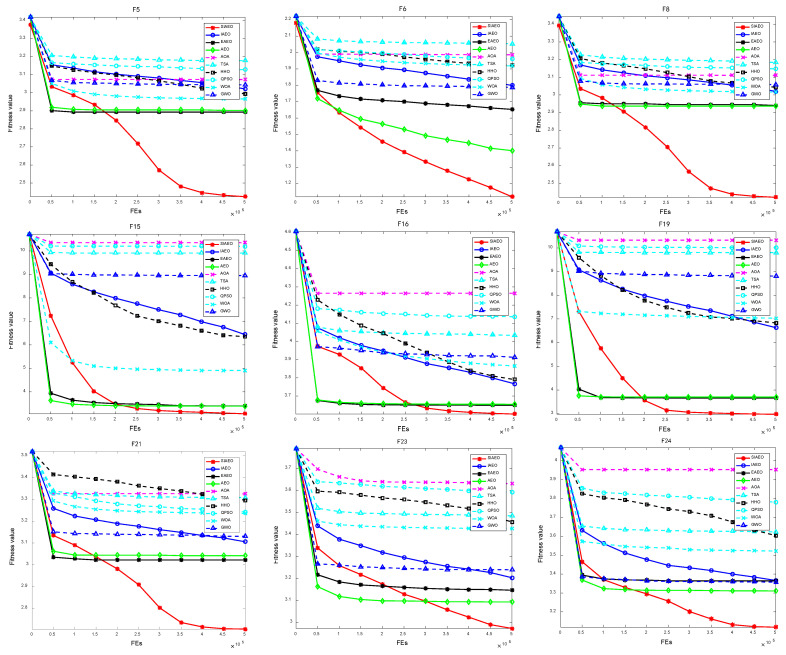
Convergence curve of nine benchmark functions of the CEC2017.

**Figure 4 biomimetics-08-00242-f004:**
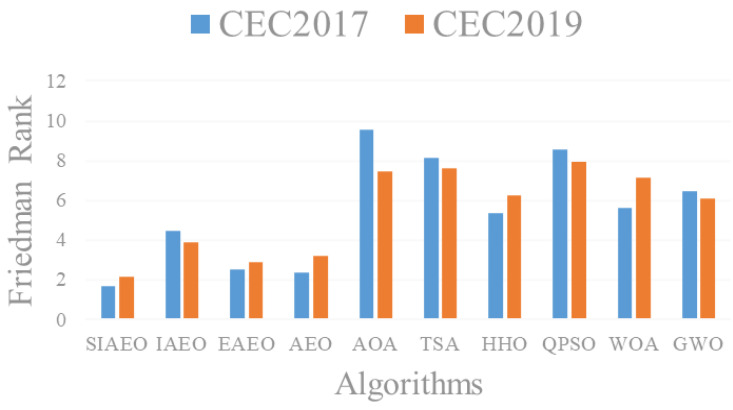
Friedman test results of the SIAEO and comparison algorithms on CEC2017 and CEC2019.

**Figure 5 biomimetics-08-00242-f005:**
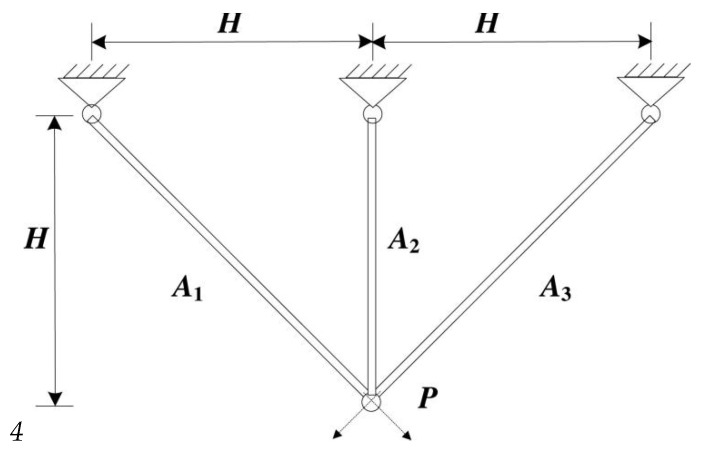
Sketch map of the three-bar truss design problem.

**Figure 6 biomimetics-08-00242-f006:**
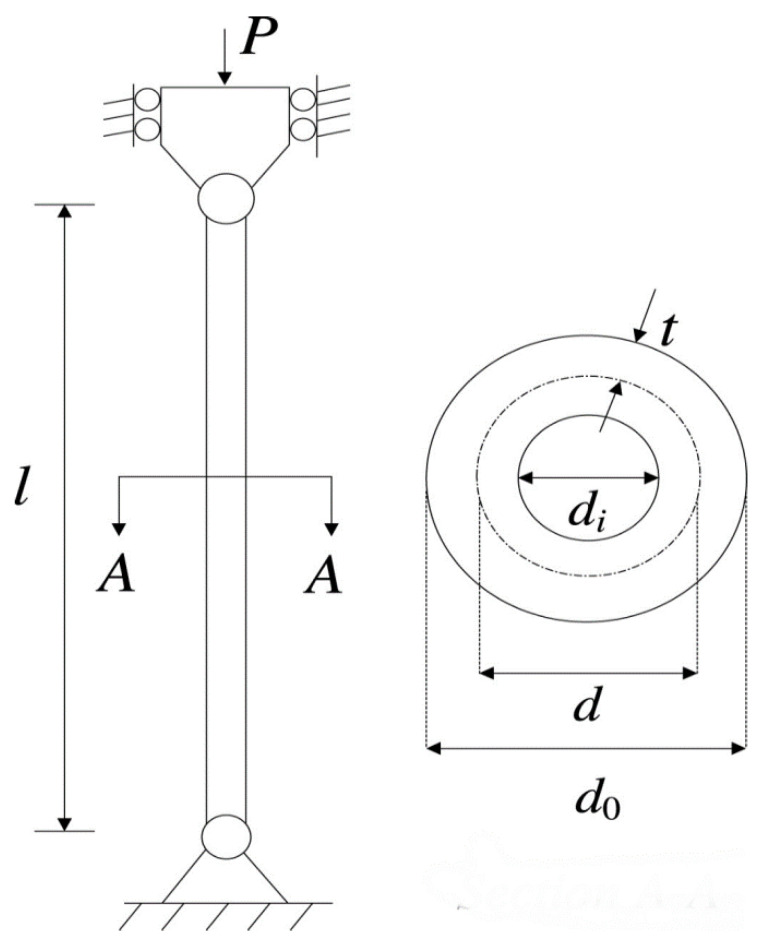
Sketch map of the tabular column design problem.

**Figure 7 biomimetics-08-00242-f007:**
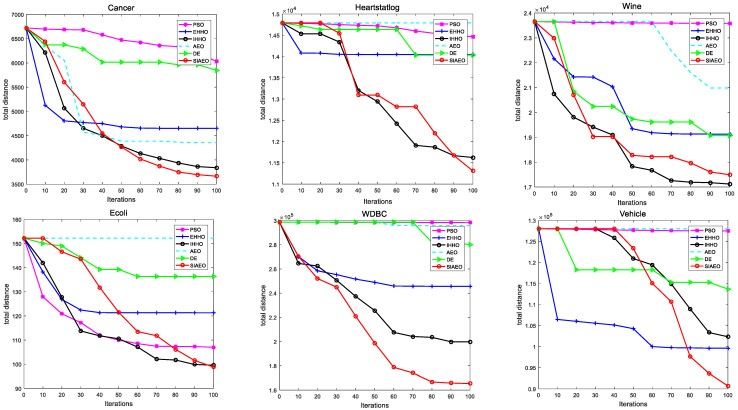
Convergence curves between the SIAEO–K-means and other algorithms on all data sets.

**Table 1 biomimetics-08-00242-t001:** CEC2019 benchmark function.

Function		Range	Dim	$F(x^{*})$
f1	Storn’s Chebyshev Polynomial Fitting Problem	[−8192, 8192]	9	1
f2	Inverse Hilbert Matrix Problem	[−16348, 16348]	16	1
f3	Lennard–Jones Minimum Energy Cluster	[−4, 4]	18	1
f4	Rastrigin’s Function	[−100, 100]	10	1
f5	Griewangk’s Function	[−100, 100]	10	1
f6	Weierstrass Function	[−100, 100]	10	1
f7	Modified Schwefel Function	[−100, 100]	10	1
f8	Expanded Schaffer F6 Function	[−100, 100]	10	1
f9	Happy Cat Function	[−100, 100]	10	1
f10	Ackley Function	[−100, 100]	10	1

**Table 2 biomimetics-08-00242-t002:** Comparison of results of SIAEO’s improved strategy on CEC2019 functions.

Functions		Algorithms				
		AEO	SAEO	CAEO	DAEO	SIAEO
f1	Ave	**0**	**0**	**0**	**0**	**0**
	Std	**0**	**0**	**0**	**0**	**0**
	Time	10 (NaN)	12 (NaN)	12 (NaN)	14 (NaN)	15
f2	Ave	3.35	3.38	**3.25**	3.28	3.36
	Std	2.68 × 10^−1^	2.65 × 10^−1^	**1.87 × 10^−2^**	1.47 × 10^−1^	2.21 × 10^−1^
	Time	10 (=)	12 (=)	13 (=)	14 (=)	17
f3	Ave	4.09 × 10^−1^	**3.58 × 10^−1^**	4.09 × 10^−1^	1.02	3.62 × 10^−1^
	Std	1.34 × 10^−15^	1.45 × 10^−1^	**6.28 × 10^−16^**	1.90	1.46 × 10^−1^
	Time	10 (−)	11 (=)	12 (−)	13 (−)	15
f4	Ave	2.00 × 10^1^	**1.13 × 10^1^**	2.54 × 10^1^	1.31 × 10^1^	1.16 × 10^1^
	Std	6.48	**2.43**	1.10 × 10^1^	6.07	3.89
	Time	11 (−)	13 (=)	13 (−)	15 (=)	18
f5	Ave	2.79 × 10^−1^	1.45 × 10^−1^	1.81 × 10^−1^	1.75 × 10^−1^	**1.10 × 10^−1^**
	Std	1.59 × 10^−1^	8.35 × 10^−2^	9.21 × 10^−2^	8.64 × 10^−2^	**5.37 × 10^−2^**
	Time	10 (−)	11 (=)	11 (−)	13 (=)	13
f6	Ave	3.09	1.25	2.19	**3.11 × 10^−1^**	4.76 × 10^−1^
	Std	2.01	1.08	1.40	5.28 × 10^−1^	**1.51 × 10^−1^**
	Time	27 (−)	28 (=)	29 (−)	30 (=)	34
f7	Ave	6.40 × 10^2^	6.74 × 10^2^	4.81 × 10^2^	6.47 × 10^2^	**4.40 × 10^2^**
	Std	**1.36 × 10^2^**	1.97 × 10^2^	2.90 × 10^2^	2.60 × 10^2^	1.43 × 10^2^
	Time	11 (−)	13 (−)	14 (=)	15 (−)	19
f8	Ave	2.47	2.78	2.44	**1.75**	2.36
	Std	**1.60 × 10^−1^**	2.71 × 10^−1^	4.72 × 10^−1^	6.22 × 10^−1^	4.77 × 10^−1^
	Time	10 (=)	12 (−)	13 (=)	14 (=)	18
f9	Ave	2.51 × 10^−1^	1.70 × 10^−1^	2.67 × 10^−1^	**1.34 × 10^−1^**	1.45 × 10^−1^
	Std	9.58 × 10^−2^	7.84 × 10^−2^	9.51 × 10^−2^	3.49 × 10^−2^	**2.54 × 10^−2^**
	Time	10 (−)	12 (=)	12 (−)	14 (=)	18
f10	Ave	1.78 × 10^1^	2.00 × 10^1^	1.29 × 10^1^	1.50 × 10^1^	**8.14**
	Std	6.25	**9.61 × 10^−4^**	9.82	9.26	7.56
	Time	10 (−)	12 (−)	13 (=)	14 (=)	17
(#)	+	7	3	5	2	
(#)	=	2	6	4	7	
(#)	−	0	0	0	0	
Friedman rank		3.434	3.139	3.095	2.639	2.419

**Table 3 biomimetics-08-00242-t003:** Algorithm parameter setting.

Algorithms	Parameter	Value
SIAEO	p	50
AOA	$C_{1},C_{2},C_{3},C_{4}$	2, 6, 2, 0.5
TSA	$p_{\max},p_{\min}$	4, 1
HHO	β	1.5
QPSO	$\omega_{\max}$, $\omega_{\min}$	1, 0.5
GWO	$a_{\max}$, $a_{\min}$	2, 0
WOA	$a_{\max}$, $a_{\min}$	2, 0

**Table 4 biomimetics-08-00242-t004:** Comparison between the SIAEO and comparison algorithm on the CEC2017 test set (*D* = 100).

Functions		Algorithms									
		SIAEO	IAEO	EAEO	AEO	AOA	TSA	HHO	QPSO	WOA	GWO
F1	Ave	1.12 × 10^4^	8.56 × 10^9^	5.90 × 10^3^	6.62 × 10^3^	2.58 × 10^11^	1.09 × 10^11^	3.49 × 10^8^	1.57 × 10^11^	1.07 × 10^7^	6.51 × 10^10^
	Std	1.03 × 10^4^	1.11 × 10^9^	4.68 × 10^3^	5.37 × 10^3^	1.71 × 10^10^	1.28 × 10^10^	3.84 × 10^7^	6.62 × 10^9^	1.81 × 10^6^	9.21 × 10^9^
	Time	34	27 (+)	28 (=)	26 (=)	124 (+)	165 (+)	26 (+)	31 (+)	82 (+)	28 (+)
F2	Ave	4.62 × 10^77^	8.67 × 10^112^	**8.53 × 10^42^**	2.14 × 10^46^	8.04 × 10^170^	5.89 × 10^140^	8.06 × 10^93^	6.30 × 10^149^	5.83 × 10^129^	2.97 × 10^132^
	Std	9.80 × 10^77^	2.19 × 10^113^	**2.34 × 10^43^**	6.04 × 10^46^	Inf	1.67 × 10^141^	2.28 × 10^94^	4.00 × 10^149^	1.65 × 10^130^	8.40 × 10^132^
	Time	34	28 (=)	29 (=)	26 (=)	125 (=)	166 (=)	27 (=)	32 (+)	83 (=)	29 (=)
F3	Ave	1.25 × 10^5^	2.81 × 10^5^	**4.81 × 10^3^**	1.07 × 10^4^	3.01 × 10^5^	2.00 × 10^5^	1.45 × 10^5^	2.69 × 10^5^	7.10 × 10^5^	2.19 × 10^5^
	Std	1.53 × 10^4^	1.40 × 10^4^	**1.40 × 10^3^**	4.61 × 10^3^	1.06 × 10^4^	2.61 × 10^4^	2.01 × 10^4^	1.23 × 10^4^	1.22 × 10^5^	1.91 × 10^4^
	Time	33	27 (+)	28 (−)	25 (−)	123 (+)	164 (+)	26 (+)	31 (+)	83 (+)	31 (+)
F4	Ave	3.14 × 10^2^	1.36 × 10^3^	**2.91 × 10^2^**	2.97 × 10^2^	8.83 × 10^4^	1.73 × 10^4^	5.94 × 10^2^	3.71 × 10^4^	5.61 × 10^2^	5.61 × 10^3^
	Std	**3.72 × 10^1^**	1.61 × 10^2^	6.11 × 10^1^	7.65 × 10^1^	1.01 × 10^4^	5.18 × 10^3^	1.23 × 10^2^	6.72 × 10^3^	8.31 × 10^1^	8.65 × 10^2^
	Time	34	27 (+)	28 (=)	25 (=)	123 (+)	163 (+)	26 (+)	31 (+)	83 (+)	31 (+)
F5	Ave	**2.64 × 10^2^**	9.40 × 10^2^	7.82 × 10^2^	7.99 × 10^2^	1.19 × 10^3^	1.50 × 10^3^	9.64 × 10^2^	1.34 × 10^3^	9.24 × 10^2^	1.11 × 10^3^
	Std	4.37 × 10^1^	6.04 × 10^1^	7.06 × 10^1^	5.26 × 10^1^	8.16 × 10^1^	9.99 × 10^1^	3.79 × 10^1^	**2.56 × 10^1^**	7.10 × 10^1^	5.93 × 10^1^
	Time	36	30 (+)	30 (+)	28 (+)	125 (+)	166 (+)	29 (+)	33 (+)	85 (+)	33 (+)
F6	Ave	**5.77**	5.74 × 10^1^	4.43 × 10^1^	2.32 × 10^1^	9.69 × 10^1^	1.13 × 10^2^	8.20 × 10^1^	8.98 × 10^1^	8.22 × 10^1^	6.12 × 10^1^
	Std	**2.09**	8.74	6.65	4.02	3.70	7.17	4.71	2.47	1.04 × 10^1^	2.57
	Time	43	40 (+)	37 (+)	35 (+)	132 (+)	173 (+)	38 (+)	40 (+)	92 (+)	40 (+)
F7	Ave	**6.27 × 10^2^**	1.60 × 10^3^	1.98 × 10^3^	2.04 × 10^3^	3.07 × 10^3^	2.83 × 10^3^	2.93 × 10^3^	2.40 × 10^3^	2.57 × 10^3^	1.74 × 10^3^
	Std	**1.10 × 10^2^**	1.85 × 10^2^	2.56 × 10^2^	2.55 × 10^2^	1.78 × 10^2^	2.05 × 10^2^	1.88 × 10^2^	9.26 × 10^1^	9.73 × 10^1^	5.81 × 10^1^
	Time	36	30 (+)	30 (+)	28 (+)	125 (+)	166 (+)	29 (+)	33 (+)	85 (+)	33 (+)
F8	Ave	**2.62 × 10^2^**	9.45 × 10^2^	8.72 × 10^2^	8.65 × 10^2^	1.29 × 10^3^	1.53 × 10^3^	1.07 × 10^3^	1.40 × 10^3^	1.04 × 10^3^	1.15 × 10^3^
	Std	4.06 × 10^1^	5.87 × 10^1^	7.96 × 10^1^	7.47 × 10^1^	1.48 × 10^2^	1.43 × 10^2^	6.84 × 10^1^	**3.04 × 10^1^**	8.37 × 10^1^	3.58 × 10^1^
	Time	36	30 (+)	30 (+)	28 (+)	125 (+)	166 (+)	29 (+)	33 (+)	85 (+)	33 (+)
F9	Ave	**1.97 × 10^3^**	4.40 × 10^4^	2.05 × 10^4^	2.21 × 10^4^	6.96 × 10^4^	1.08 × 10^5^	4.04 × 10^4^	5.64 × 10^4^	3.59 × 10^4^	4.09 × 10^4^
	Std	**9.92 × 10^2^**	6.36 × 10^3^	2.11 × 10^3^	2.73 × 10^3^	3.26 × 10^3^	2.26 × 10^4^	2.41 × 10^3^	3.10 × 10^3^	1.05 × 10^4^	4.44 × 10^3^
	Time	36	30 (+)	30 (+)	28 (+)	124 (+)	166 (+)	29 (+)	33 (+)	85 (+)	33 (+)
F10	Ave	**1.27 × 10^4^**	1.95 × 10^4^	1.43 × 10^4^	1.48 × 10^4^	2.89 × 10^4^	2.63 × 10^4^	1.87 × 10^4^	2.94 × 10^4^	1.96 × 10^4^	2.86 × 10^4^
	Std	1.05 × 10^3^	1.69 × 10^3^	1.44 × 10^3^	1.09 × 10^3^	8.93 × 10^2^	1.34 × 10^3^	1.48 × 10^3^	1.25 × 10^3^	2.48 × 10^3^	**7.18 × 10^2^**
	Time	38	33 (+)	32 (+)	30 (+)	127 (+)	167 (+)	32 (+)	35 (+)	87 (+)	35 (+)
F11	Ave	**8.33 × 10^2^**	2.38 × 10^4^	1.08 × 10^3^	9.27 × 10^2^	1.40 × 10^5^	7.28 × 10^4^	2.36 × 10^3^	9.15 × 10^4^	4.61 × 10^3^	5.17 × 10^4^
	Std	**1.36 × 10^2^**	4.98 × 10^3^	1.84 × 10^2^	1.50 × 10^2^	1.41 × 10^4^	1.41 × 10^4^	3.67 × 10^2^	2.34 × 10^3^	8.23 × 10^2^	1.07 × 10^4^
	Time	34	28 (+)	29 (+)	26 (=)	124 (+)	164 (+)	27 (=)	32 (+)	84 (+)	32 (+)
F12	Ave	2.18 × 10^7^	6.97 × 10^8^	2.83 × 10^6^	**2.59 × 10^6^**	1.85 × 10^11^	5.52 × 10^10^	4.16 × 10^8^	1.02 × 10^11^	6.44 × 10^8^	1.89 × 10^10^
	Std	8.83 × 10^6^	5.07 × 10^7^	1.23 × 10^6^	**1.14 × 10^6^**	1.56 × 10^10^	1.55 × 10^10^	1.95 × 10^8^	8.02 × 10^9^	2.72 × 10^8^	2.33 × 10^9^
	Time	35	31 (+)	30 (−)	28 (−)	125 (+)	166 (+)	29 (+)	33 (+)	86 (+)	33 (+)
F13	Ave	**2.76 × 10^3^**	1.04 × 10^7^	1.11 × 10^4^	4.24 × 10^3^	4.30 × 10^10^	1.46 × 10^10^	5.64 × 10^6^	2.79 × 10^10^	8.92 × 10^4^	2.89 × 10^9^
	Std	**1.45 × 10^3^**	3.88 × 10^6^	5.21 × 10^3^	2.95 × 10^3^	3.95 × 10^9^	5.36 × 10^9^	1.48 × 10^6^	1.69 × 10^8^	3.95 × 10^4^	9.55 × 10^8^
	Time	34	29 (+)	29 (+)	27 (=)	123 (+)	164 (+)	28 (+)	32 (+)	84 (+)	32 (+)
F14	Ave	8.20 × 10^4^	2.84 × 10^6^	**3.95 × 10^4^**	5.87 × 10^4^	4.36 × 10^7^	6.45 × 10^6^	1.34 × 10^6^	1.18 × 10^7^	1.21 × 10^6^	8.47 × 10^6^
	Std	5.17 × 10^4^	9.97 × 10^5^	**1.57 × 10^4^**	3.12 × 10^4^	1.70 × 10^7^	4.70 × 10^6^	2.10 × 10^5^	1.43 × 10^6^	4.81 × 10^5^	3.47 × 10^6^
	Time	37	33 (+)	32 (=)	30 (=)	127 (+)	167 (+)	31 (+)	35 (+)	87 (+)	35 (+)
F15	Ave	**9.44 × 10^2^**	5.96 × 10^5^	2.48 × 10^3^	2.46 × 10^3^	2.24 × 10^10^	8.01 × 10^9^	1.49 × 10^6^	1.55 × 10^10^	8.04 × 10^4^	8.42 × 10^8^
	Std	**9.83 × 10^2^**	1.97 × 10^5^	2.17 × 10^3^	2.93 × 10^3^	3.29 × 10^9^	4.85 × 10^9^	3.45 × 10^5^	1.17 × 10^9^	4.73 × 10^4^	1.90 × 10^8^
	Time	34	28 (+)	29 (+)	26 (=)	123 (+)	164 (+)	27 (+)	32 (+)	84 (+)	32 (+)
F16	Ave	**3.90 × 10^3^**	5.09 × 10^3^	4.46 × 10^3^	4.51 × 10^3^	1.84 × 10^4^	1.08 × 10^4^	5.86 × 10^3^	1.37 × 10^4^	7.20 × 10^3^	8.15 × 10^3^
	Std	5.91 × 10^2^	**4.47 × 10^2^**	6.25 × 10^2^	8.49 × 10^2^	2.06 × 10^3^	1.87 × 10^3^	7.92 × 10^2^	1.17 × 10^3^	1.42 × 10^3^	7.17 × 10^2^
	Time	35	30 (+)	30 (+)	27 (=)	124 (+)	165 (+)	28 (+)	33 (+)	85 (+)	33 (+)
F17	Ave	**2.51 × 10^3^**	3.74 × 10^3^	3.70 × 10^3^	3.50 × 10^3^	4.28 × 10^6^	1.11 × 10^5^	4.70 × 10^3^	5.59 × 10^4^	4.92 × 10^3^	6.42 × 10^3^
	Std	4.75 × 10^2^	**4.15 × 10^2^**	6.34 × 10^2^	6.24 × 10^2^	2.73 × 10^6^	1.14 × 10^5^	5.67 × 10^2^	6.07 × 10^3^	4.37 × 10^2^	5.28 × 10^2^
	Time	40	37 (+)	35 (+)	33 (+)	130 (+)	171 (+)	35 (+)	39 (+)	91 (+)	38 (+)
F18	Ave	2.26 × 10^5^	4.19 × 10^6^	1.72 × 10^5^	**1.58 × 10^5^**	5.85 × 10^7^	7.24 × 10^6^	3.18 × 10^6^	2.25 × 10^7^	1.78 × 10^6^	1.15 × 10^7^
	Std	1.46 × 10^5^	1.33 × 10^6^	**5.65 × 10^4^**	7.04 × 10^4^	2.21 × 10^7^	3.98 × 10^6^	1.18 × 10^6^	1.13 × 10^7^	5.22 × 10^5^	3.57 × 10^6^
	Time	35	30 (+)	30 (=)	27 (=)	124 (+)	165 (+)	28 (+)	33 (+)	85 (=)	33 (+)
F19	Ave	**8.99 × 10^2^**	1.22 × 10^6^	4.49 × 10^3^	5.03 × 10^3^	2.12 × 10^10^	6.43 × 10^9^	4.93 × 10^6^	1.04 × 10^10^	1.03 × 10^7^	6.47 × 10^8^
	Std	**7.49 × 10^2^**	3.79 × 10^5^	6.07 × 10^3^	4.33 × 10^3^	2.48 × 10^9^	5.20 × 10^9^	1.30 × 10^6^	2.63 × 10^9^	7.15 × 10^6^	1.20 × 10^8^
	Time	71	78 (+)	66 (+)	64 (+)	161 (+)	202 (+)	73 (+)	69 (+)	121 (+)	69 (+)
F20	Ave	**2.22 × 10^3^**	3.04 × 10^3^	3.46 × 10^3^	3.20 × 10^3^	5.30 × 10^3^	4.24 × 10^3^	3.88 × 10^3^	4.71 × 10^3^	4.05 × 10^3^	4.35 × 10^3^
	Std	**3.23 × 10^2^**	5.23 × 10^2^	3.70 × 10^2^	7.41 × 10^2^	3.25 × 10^2^	3.90 × 10^2^	4.55 × 10^2^	3.64 × 10^2^	4.24 × 10^2^	6.19 × 10^2^
	Time	42	39 (+)	37 (+)	34 (+)	131 (+)	172 (+)	38 (+)	40 (+)	92 (+)	40 (+)
F21	Ave	**5.04 × 10^2^**	1.15 × 10^3^	1.05 × 10^3^	1.10 × 10^3^	2.12 × 10^3^	2.03 × 10^3^	1.91 × 10^3^	1.74 × 10^3^	1.72 × 10^3^	1.34 × 10^3^
	Std	4.17 × 10^1^	8.51 × 10^1^	8.91 × 10^1^	1.49 × 10^2^	1.68 × 10^2^	1.67 × 10^2^	1.52 × 10^2^	**1.90 × 10^1^**	2.20 × 10^2^	5.08 × 10^1^
	Time	65	69 (+)	59 (+)	57 (+)	153 (+)	194 (+)	64 (+)	62 (+)	114 (+)	62 (+)
F22	Ave	**1.46 × 10^4^**	2.19 × 10^4^	1.71 × 10^4^	1.59 × 10^4^	3.12 × 10^4^	2.83 × 10^4^	2.09 × 10^4^	2.75 × 10^4^	2.09 × 10^4^	2.99 × 10^4^
	Std	1.55 × 10^3^	**7.94 × 10^2^**	1.30 × 10^3^	2.02 × 10^3^	1.02 × 10^3^	1.61 × 10^3^	1.66 × 10^3^	3.36 × 10^3^	1.76 × 10^3^	8.82 × 10^2^
	Time	68	72 (+)	62 (+)	59 (=)	157 (+)	197 (+)	68 (+)	65 (+)	117 (+)	65 (+)
F23	Ave	**9.05 × 10^2^**	1.42 × 10^3^	1.38 × 10^3^	1.24 × 10^3^	4.26 × 10^3^	3.03 × 10^3^	2.65 × 10^3^	3.89 × 10^3^	2.66 × 10^3^	1.72 × 10^3^
	Std	**5.75 × 10^1^**	1.02 × 10^2^	1.78 × 10^2^	6.39 × 10^1^	3.66 × 10^2^	3.31 × 10^2^	2.83 × 10^2^	7.78 × 10^1^	2.87 × 10^2^	6.67 × 10^1^
	Time	79	88 (+)	74 (+)	71 (+)	168 (+)	209 (+)	80 (+)	76 (+)	129 (+)	76 (+)
F24	Ave	**1.31 × 10^3^**	2.13 × 10^3^	2.31 × 10^3^	2.05 × 10^3^	8.98 × 10^3^	4.19 × 10^3^	3.66 × 10^3^	6.00 × 10^3^	3.32 × 10^3^	2.27 × 10^3^
	Std	**1.04 × 10^2^**	9.40 × 10^1^	2.53 × 10^2^	9.81 × 10^1^	1.69 × 10^3^	2.47 × 10^2^	3.54 × 10^2^	1.65 × 10^2^	2.94 × 10^2^	7.89 × 10^1^
	Time	74	83 (+)	69 (+)	67 (+)	156 (+)	192 (+)	75 (+)	71 (+)	119 (+)	72 (+)
F25	Ave	8.98 × 10^2^	1.84 × 10^3^	**7.95 × 10^2^**	8.11 × 10^2^	2.53 × 10^4^	8.41 × 10^3^	1.12 × 10^3^	1.16 × 10^4^	1.03 × 10^3^	4.87 × 10^3^
	Std	5.61 × 10^1^	1.22 × 10^2^	**4.70 × 10^1^**	7.14 × 10^1^	1.95 × 10^3^	1.59 × 10^3^	5.81 × 10^1^	2.05 × 10^2^	5.37 × 10^1^	5.55 × 10^2^
	Time	79	89 (+)	73 (−)	71 (=)	160 (+)	196 (+)	81 (+)	76 (+)	124 (+)	76 (+)
F26	Ave	**8.64 × 10^3^**	1.25 × 10^4^	1.72 × 10^4^	1.56 × 10^4^	4.95 × 10^4^	2.96 × 10^4^	2.18 × 10^4^	2.62 × 10^4^	2.74 × 10^4^	1.77 × 10^4^
	Std	**8.66 × 10^2^**	8.07 × 10^3^	6.07 × 10^3^	6.34 × 10^3^	2.29 × 10^3^	1.83 × 10^3^	2.67 × 10^3^	2.87 × 10^2^	3.34 × 10^3^	1.11 × 10^3^
	Time	86	97 (+)	80 (+)	78 (+)	167 (+)	203 (+)	89 (+)	83 (+)	131 (+)	83 (+)
F27	Ave	**8.97 × 10^2^**	1.29 × 10^3^	1.39 × 10^3^	1.31 × 10^3^	8.13 × 10^3^	3.26 × 10^3^	1.76 × 10^3^	5.26 × 10^3^	2.57 × 10^3^	1.71 × 10^3^
	Std	**5.36 × 10^1^**	1.00 × 10^2^	1.46 × 10^2^	2.58 × 10^2^	4.21 × 10^3^	6.08 × 10^2^	4.70 × 10^2^	1.32 × 10^2^	1.02 × 10^3^	1.66 × 10^2^
	Time	97	114 (+)	92 (+)	90 (+)	179 (+)	215 (+)	102 (+)	95 (+)	143 (+)	95 (+)
F28	Ave	7.63 × 10^2^	1.96 × 10^3^	6.55 × 10^2^	**6.48 × 10^2^**	3.28 × 10^4^	1.31 × 10^4^	8.43 × 10^2^	1.05 × 10^4^	8.17 × 10^2^	6.44 × 10^3^
	Std	4.63 × 10^1^	1.77 × 10^2^	2.89 × 10^1^	**2.68 × 10^1^**	2.76 × 10^3^	3.27 × 10^3^	3.61 × 10^1^	2.50 × 10^2^	3.41 × 10^1^	7.42 × 10^2^
	Time	92	107 (+)	87 (−)	85 (−)	174 (+)	209 (+)	98 (+)	90 (+)	138 (+)	90 (+)
F29	Ave	**2.94 × 10^3^**	4.84 × 10^3^	4.40 × 10^3^	4.07 × 10^3^	2.62 × 10^5^	1.48 × 10^4^	6.15 × 10^3^	6.21 × 10^4^	1.07 × 10^4^	7.88 × 10^3^
	Std	4.85 × 10^2^	4.31 × 10^2^	**3.50 × 10^2^**	5.72 × 10^2^	1.23 × 10^5^	5.60 × 10^3^	3.93 × 10^2^	2.89 × 10^3^	3.23 × 10^3^	5.99 × 10^2^
	Time	61	67 (+)	56 (+)	54 (+)	144 (+)	179 (+)	61 (+)	59 (+)	107 (+)	59 (+)
F30	Ave	**1.83 × 10^4^**	2.65 × 10^7^	2.32 × 10^4^	1.90 × 10^4^	3.69 × 10^10^	1.21 × 10^10^	3.55 × 10^7^	1.93 × 10^10^	1.77 × 10^8^	2.45 × 10^9^
	Std	**8.12 × 10^3^**	6.51 × 10^6^	1.49 × 10^4^	8.99 × 10^3^	6.59 × 10^9^	4.17 × 10^9^	9.20 × 10^6^	1.92 × 10^9^	5.02 × 10^7^	6.43 × 10^8^
	Time	88	103 (+)	84 (=)	81 (=)	188 (+)	229 (+)	106 (+)	97 (+)	150 (+)	96 (+)
(#)	best	21	0	6	3	0	0	0	0	0	0
(#)	+		29	20	16	30	30	28	30	28	30
(#)	=		1	6	11	0	0	2	0	2	0
(#)	−		0	4	3	0	0	0	0	0	0
Friedman rank		1.69	4.45	2.52	2.38	9.57	8.12	5.34	8.54	5.63	6.47

**Table 5 biomimetics-08-00242-t005:** Comparison between the SIAEO and comparison algorithm on the CEC2019 test set.

Functions		Algorithm									
		SIAEO	IAEO	EAEO	AEO	AOA	TSA	HHO	QPSO	WOA	GWO
f1	Ave	**0.00**	**0.00**	**0.00**	**0.00**	3.75 × 10^−15^	1.78 × 10^2^	**0.00**	1.16 × 10^−9^	3.94 × 10^6^	4.02 × 10^4^
	Std	**0.00**	**0.00**	**0.00**	**0.00**	1.19 × 10^−14^	3.97 × 10^2^	**0.00**	2.41 × 10^−9^	5.26 × 10^6^	1.06 × 10^5^
	Time	1.71	0.95 (=)	1.26 (=)	1.08 (=)	1.34 (=)	1.71 (+)	0.96 (=)	1.47 (=)	1.18 (+)	0.78 (+)
f2	Ave	3.43	**3.31**	3.55	3.37	3.90	5.58 × 10^2^	3.98	4.00	6.60 × 10^3^	7.05 × 10^2^
	Std	1.70 × 10^−1^	**3.39 × 10^−2^**	3.18 × 10^−1^	2.35 × 10^−1^	1.89 × 10^−1^	1.67 × 10^2^	6.51 × 10^−2^	4.28 × 10^−4^	6.17 × 10^2^	3.85 × 10^2^
	Time	1.57	0.82 (=)	1.18 (=)	1.00 (=)	1.84 (+)	2.46 (+)	0.85 (+)	1.39 (+)	1.42 (+)	0.72 (+)
f3	Ave	**3.73 × 10^−1^**	3.14	4.09 × 10^−1^	4.38 × 10^−1^	3.82	7.68	2.75	5.77	2.90	3.89
	Std	**1.31 × 10^−1^**	1.02	2.41 × 10^−8^	9.17 × 10^−2^	5.37 × 10^−1^	2.27	1.22	1.05	2.59	4.41 × 10^−1^
	Time	1.65	0.81 (+)	1.18 (=)	0.98 (=)	2.01 (+)	2.61 (+)	0.83 (+)	1.42 (+)	1.47 (+)	0.72 (+)
f4	Ave	**1.55 × 10^1^**	1.80 × 10^1^	1.69 × 10^1^	1.69 × 10^1^	5.20 × 10^1^	6.34 × 10^1^	4.22 × 10^1^	6.07 × 10^1^	5.19 × 10^1^	2.81 × 10^1^
	Std	3.17	6.95	5.31	6.76	7.68	1.07 × 10^1^	8.57	4.67	1.99 × 10^1^	**2.35**
	Time	1.48	0.82 (=)	1.16 (=)	0.98 (=)	1.33 (+)	1.72 (+)	0.86 (+)	1.38 (+)	1.12 (+)	0.68 (+)
f5	Ave	**2.07 × 10^−1^**	9.30 × 10^−1^	2.10 × 10^−1^	2.73 × 10^−1^	4.35 × 10^1^	4.93	9.67 × 10^−1^	5.22 × 10^1^	7.94 × 10^−1^	2.60
	Std	1.72 × 10^−1^	1.39 × 10^−1^	**5.42 × 10^−2^**	1.31 × 10^−1^	9.09	3.62	3.39 × 10^−1^	9.89	5.18 × 10^−1^	6.25 × 10^−1^
	Time	1.72	0.84 (+)	1.17 (=)	0.99 (=)	1.34 (+)	1.73 (+)	0.86 (+)	1.38 (+)	1.13 (+)	0.69 (+)
f6	Ave	**1.09**	1.68	2.59	3.32	7.30	3.79	5.35	6.62	6.23	3.22
	Std	7.40 × 10^−1^	2.78 × 10^−1^	6.61 × 10^−1^	1.63	8.01 × 10^−1^	8.79 × 10^−1^	8.07 × 10^−1^	5.26 × 10^−1^	1.24	**2.76 × 10^−1^**
	Time	3.29	3.04 (+)	2.83 (+)	2.64 (+)	2.97 (+)	3.36 (+)	2.85 (+)	3.03 (+)	2.77 (+)	2.33 (+)
f7	Ave	**5.75 × 10^2^**	7.65 × 10^2^	7.43 × 10^2^	1.05 × 10^3^	1.47 × 10^3^	1.62 × 10^3^	1.27 × 10^3^	1.57 × 10^3^	1.48 × 10^3^	1.09 × 10^3^
	Std	2.36 × 10^2^	2.88 × 10^2^	2.93 × 10^2^	3.09 × 10^2^	2.13 × 10^2^	**1.36 × 10^2^**	3.96 × 10^2^	2.68 × 10^2^	3.44 × 10^2^	2.29 × 10^2^
	Time	1.37	0.83 (=)	1.16 (=)	0.98 (+)	1.32 (+)	1.70 (+)	0.88 (+)	1.36 (+)	1.11 (+)	0.69 (+)
f8	Ave	**2.73**	2.92	2.89	2.90	3.47	3.09	3.54	3.36	3.28	2.96
	Std	2.83 × 10^−1^	3.03 × 10^−1^	3.20 × 10^−1^	3.72 × 10^−1^	1.44 × 10^−1^	2.22 × 10^−1^	1.24 × 10^−1^	**1.13 × 10^−1^**	2.53 × 10^−1^	2.29 × 10^−1^
	Time	1.32	0.81 (=)	1.13 (=)	0.96 (=)	1.31 (+)	1.68 (+)	0.87 (+)	1.34 (+)	1.10 (=)	0.67 (=)
f9	Ave	**2.16 × 10^−1^**	2.68 × 10^−1^	2.74 × 10^−1^	3.41 × 10^−1^	1.73	3.88 × 10^−1^	5.99 × 10^−1^	6.53 × 10^−1^	5.88 × 10^−1^	2.55 × 10^−1^
	Std	**3.05 × 10^−2^**	6.60 × 10^−2^	1.11 × 10^−1^	9.51 × 10^−2^	4.13 × 10^−1^	1.18 × 10^−1^	1.18 × 10^−1^	9.44 × 10^−2^	1.50 × 10^−1^	8.00 × 10^−2^
	Time	1.30	0.78 (=)	1.12 (=)	0.94 (+)	1.28 (+)	1.67 (+)	0.82 (+)	1.32 (+)	1.08 (+)	0.65 (=)
f10	Ave	**1.50 × 10^1^**	1.58 × 10^1^	1.68 × 10^1^	1.80 × 10^1^	2.03 × 10^1^	2.04 × 10^1^	2.02 × 10^1^	2.03 × 10^1^	2.00 × 10^1^	2.04 × 10^1^
	Std	8.54	7.23	6.85	6.32	**3.83 × 10^−2^**	1.09 × 10^−1^	1.23 × 10^−1^	1.47 × 10^−1^	4.14 × 10^−2^	5.64 × 10^−2^
	Time	1.36	0.81 (=)	1.14 (=)	0.96 (=)	1.31 (+)	1.68 (+)	0.87 (+)	1.34 (+)	1.10 (+)	0.67 (+)
(#)	best	9	2	1	1	0	0	1	0	0	0
(#)	+		3	1	3	9	10	9	9	9	8
(#)	=		7	9	7	1	0	1	1	1	2
(#)	−		0	0	0	0	0	0	0	0	0
Friedman rank		2.16	3.88	2.88	3.21	7.46	7.61	6.26	7.93	7.16	6.12

**Table 6 biomimetics-08-00242-t006:** The CP values of the SIAEO and comparison algorithms on CEC2017 and CEC2019 test sets.

SIAEO. VS	CEC2017 (D = 100)			CEC2019		
	(#)+	(#)−	CP	(#)+	(#)−	CP
IAEO	29	0	29	3	0	3
EAEO	20	4	16	1	0	1
AEO	16	3	13	3	0	3
AOA	30	0	30	9	0	9
TSA	30	0	30	10	0	10
HHO	28	0	28	9	0	9
QPSO	30	0	30	9	0	9
WOA	28	0	28	9	0	9
GWO	30	0	30	8	0	8

**Table 7 biomimetics-08-00242-t007:** Wilcoxon sign rank test results of the SIAEO and comparison algorithms at 0.05 significance level.

SIAEO. VS	CEC2017 (D = 100)				CEC2019			
	R+	R−	*p*	Significance	R+	R−	*p*	Significance
IAEO	465	0	0.0000	+	43	2	0.0117	+
EAEO	295	170	0.1986	=	45	0	0.0039	+
AEO	293	172	0.2134	=	44	1	0.0078	+
AOA	465	0	0.0000	+	55	0	0.0020	+
TSA	465	0	0.0000	+	55	0	0.0020	+
HHO	465	0	0.0000	+	45	0	0.0039	+
QPSO	465	0	0.0000	+	55	0	0.0020	+
WOA	465	0	0.0000	+	55	0	0.0020	+
GWO	465	0	0.0000	+	55	0	0.0020	+

**Table 8 biomimetics-08-00242-t008:** Optimal results of various methods for the Three bar truss design problem.

Algorithm	The Best Decision Variables		Minimum Cost
	$x_{1}$	$x_{2}$	
CS [5]	0.78867	0.40902	263.9716
CSA [36]	0.788638976	0.408350573	263.895844
SSA [9]	0.78866541	0.408275784	263.89584
Ray and Sain [37]	0.795	0.395	264.3
MBA [38]	0.7885650	0.4085597	263.89585
PHSSA [39]	0.82299	0.31925	264.701723
SIAEO	**0.788675136**	**0.40824828**	**263.895843**

**Table 9 biomimetics-08-00242-t009:** Optimal results of various methods for the Himmelblau’s nonlinear problems.

Algorithm	The Best Decision Variables					Minimum Cost
	$x_{1}$	$x_{2}$	$x_{3}$	$x_{4}$	$x_{5}$	
Himmelblau [40]	N/A	N/A	N/A	N/A	N/A	−30,373.949
Deb [41]	N/A	N/A	N/A	N/A	N/A	−30,665.539
He et al. [42]	78	33	29.995256	45	36.775813	−30,665.539
Dimopoulos [43]	78	33	29.995256	45	36.775813	−30,665.54
CS [5]	78	33	29.99616	45	36.77605	−30,665.233
CSA [36]	78	33	29.995256	45	36.775813	−30,665.53867
SIAEO	**78**	**33**	**29.995256**	**45**	**36.775812**	**−30,665.53867**

**Table 10 biomimetics-08-00242-t010:** Optimal results of each algorithm for the tabular column design.

Algorithm	The Best Decision Variables		Minimum Cost
	$x_{1}$	$x_{2}$	
CS [5]	5.45139	0.29196	26.53217
CSA [36]	5.45116	0.29196	26.5313
Hsu and Liu [45]	5.4507	0.292	25.5316
Rao [44]	5.44	0.293	26.5323
SIAEO	**5.4512**	**0.29167**	**26.526**

**Table 11 biomimetics-08-00242-t011:** Optimal results of various methods for the Gas transmission compressor design problem.

Algorithm	The Best Decision Variables				Minimum Cost
	$x_{1}$	$x_{2}$	$x_{3}$	$x_{4}$	
WOA [31]	50	1.18	24.58	0.3883	2,964,900
SSA [9]	26.19	1.10	21.47	0.2119	3,034,100
BOA [47]	33.19	1.10	26.48	0.2162	3,007,000
SOS [48]	50	1.18	24.58	0.3883	2,964,900
PSO [49]	31.79	1.10	31.57	0.2224	3,050,900
DE [50]	50	1.18	24.59	0.3884	2,964,900
SIAEO	**50**	**1.17828**	**24.59259**	**0.38835**	**2,964,895**

**Table 12 biomimetics-08-00242-t012:** Characteristics of the nine datasets.

Datasets	Instances	Classes	Features	Dimension
Cancer	683	2	9	18
Heartstatlog	270	2	13	26
Wine	178	3	13	39
Ecoli	336	8	7	56
WDBC	569	2	30	60
Vehicle	846	4	18	72
Segmentation	210	7	18	126
Air	359	3	64	192
Abalone	4177	29	7	203

**Table 13 biomimetics-08-00242-t013:** Comparison of K-means optimization results of six algorithms on nine datasets.

Algorithm		SIAEO- K-Means	IAEO- K-Means	EAEO- K-Means	AEO- K-Means	DE-K-Means	PSO- K-Means
Cancer	mean	**3.67 × 10^3^**	4.65 × 10^3^	3.84 × 10^3^	4.36 × 10^3^	5.85 × 10^3^	6.03 × 10^3^
	Std	2.94 × 10^2^	**1.59**	3.23 × 10^2^	2.99 × 10^2^	4.01 × 10^2^	4.32 × 10^2^
	Rank	1	4	2	3	5	6
Heartstatlog	mean	**1.13 × 10^4^**	1.40 × 10^4^	1.16 × 10^4^	1.48 × 10^4^	1.40 × 10^4^	1.45 × 10^4^
	Std	7.02 × 10^1^	1.60 × 10^2^	5.32 × 10^2^	**0.00**	6.37 × 10^2^	1.94 × 10^1^
	Rank	1	3	2	6	4	5
Wine	mean	1.75 × 10^4^	1.91 × 10^4^	**1.71 × 10^4^**	2.10 × 10^4^	1.91 × 10^4^	2.36 × 10^4^
	Std	1.26 × 10^3^	1.05 × 10^3^	2.42 × 10^2^	1.94 × 10^3^	8.75 × 10^2^	**7.46 × 10^1^**
	Rank	2	3	1	5	4	6
Ecoli	mean	**9.88 × 10^1^**	1.21 × 10^2^	9.96 × 10^1^	1.52 × 10^2^	1.36 × 10^2^	1.07 × 10^2^
	Std	3.63	1.17 × 10^1^	1.04 × 10^1^	**0.00**	8.32	4.21
	Rank	1	4	2	6	5	3
WDBC	mean	**1.65 × 10^5^**	2.46 × 10^5^	2.00 × 10^5^	2.96 × 10^5^	2.80 × 10^5^	2.98 × 10^5^
	Std	4.22 × 10^3^	3.20 × 10^1^	3.96 × 10^4^	3.98 × 10^3^	2.07 × 10^4^	**2.17 × 10^1^**
	Rank	1	3	2	5	4	6
Vehicle	mean	**9.07 × 10^4^**	9.96 × 10^4^	1.02 × 10^5^	1.28 × 10^5^	1.14 × 10^5^	1.27 × 10^5^
	Std	7.10 × 10^3^	5.36 × 10^3^	2.97 × 10^3^	**0.00**	7.32 × 10^3^	1.40 × 10^1^
	Rank	1	2	3	6	4	5
Segmentation	mean	**2.90 × 10^4^**	3.53 × 10^4^	3.08 × 10^4^	3.56 × 10^4^	3.66 × 10^4^	3.91 × 10^4^
	Std	9.79 × 10^2^	5.38 × 10^2^	2.66 × 10^2^	5.05 × 10^3^	1.33 × 10^3^	**1.50 × 10^2^**
	Rank	1	3	2	4	5	6
air	mean	4.48 × 10^1^	**3.57 × 10^1^**	3.97 × 10^1^	7.72 × 10^1^	1.35 × 10^2^	6.22 × 10^1^
	Std	3.92	2.71	6.68 × 10^−1^	1.81	**0.00**	7.03
	Rank	3	1	2	5	6	4
abalone	mean	**1.17 × 10^3^**	1.59 × 10^3^	1.27 × 10^3^	1.91 × 10^3^	1.91 × 10^3^	1.58 × 10^3^
	Std	**6.17**	4.60 × 10^1^	9.40 × 10^1^	6.98 × 10^1^	3.51 × 10^1^	1.87 × 10^2^
	Rank	1	4	2	5	6	3
count		7	1	1	0	0	0
Avg_Rank		1.33	3.00	2.00	5.00	4.78	4.89
Total Rank		1	3	2	6	4	5

## Data Availability

This study did not conduct data archiving, as the algorithm has randomness, and the results obtained each time are different.

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
