# Peer review of "Improved Environmental Stimulus and Biological Competition Tactics Interactive Artificial Ecological Optimization Algorithm for Clustering"

_biomimetics, 2023, doi:10.3390/biomimetics8020242_

Round 1

Reviewer 1 Report

The article describes the development of an interactive artificial ecological optimization algorithm (SIAEO) that aims to overcome the problem of getting stuck in local optima that often occurs in traditional ecological optimization algorithms. SIAEO achieves this by incorporating population diversity and competition mechanisms, and by introducing a stochastic mean suppression alternation exploitation process to help the algorithm escape local optima. The effectiveness of SIAEO is compared to other improved algorithms using the CEC2017 and CEC2019 test sets.

The article is well presented, though it would have been interesting to compare the results of the proposed algorithm also with respect to other standard black-box optimization techniques such as Differential Evolution, Particle Swarm Optimization and others.

With this regard, I suggest the authors the following references where a sound comparison among different metaheuristics has been performed (even though on different problems):

- Agresta, A. et al. (2023, April). An intelligent optimised estimation of the hydraulic jump roller length. In Applications of Evolutionary Computation: 26th European Conference, EvoApplications 2023, Held as Part of EvoStar 2023, Brno, Czech Republic, April 12–14, 2023, Proceedings (pp. 475-490). Cham: Springer Nature Switzerland.

- Yeoh, J. M. et al. (2019). A clustering system for dynamic data streams based on metaheuristic optimisation. Mathematics, 7(12), 1229.

Author Response

  In response to Reviewer 1's comments, we have provided an overview of the work of these two papers at the end of the article and identified them as the main direction for our future in-depth research on AEO algorithms.

  Revisions can be found in the last paragraph of Part 5 on page 27 and references [54] and [55] on page 29 of the text.

Reviewer 2 Report

1. Before describing the optimization algorithm, it is recommended to formulate the problem statement, describing the class of problems to be solved and explaining the designations fit , upj, lowj.

2. Conditions (12) agree with those used in the description of the pseudocode (clarify the inequality sign).

Author Response

  In response to Reviewer 2's comments,1) We have added a description of the optimization problem and explained the meanings of   , , and fit(), as shown on pages 3 and 6 of the text.2) Modified the inconsistency between line 12 of pseudocode on page 7 and formula (12).

Round 2

Reviewer 1 Report

The authors have addressed my previous comments.